# The deubiquitinase Ubp3/Usp10 constrains glucose-mediated mitochondrial repression via phosphate budgeting

Vineeth Vengayil[1,2], Shreyas Niphadkar[1,2], Swagata Adhikary[1,2], Sriram Varahan[1], Sunil Laxman[1]*

[1]Institute for Stem Cell Science and Regenerative Medicine (DBT-inStem), Bangalore, India; [2]Manipal Academy of Higher Education, Bangalore, India

*For correspondence:
sunil.laxman@gmail.com

Competing interest: The authors declare that no competing interests exist.

**Abstract** Many cells in high glucose repress mitochondrial respiration, as observed in the Crabtree and Warburg effects. Our understanding of biochemical constraints for mitochondrial activation is limited. Using a *Saccharomyces cerevisiae* screen, we identified the conserved deubiquitinase Ubp3 (Usp10), as necessary for mitochondrial repression. Ubp3 mutants have increased mitochondrial activity despite abundant glucose, along with decreased glycolytic enzymes, and a rewired glucose metabolic network with increased trehalose production. Utilizing *Δubp3* cells, along with orthogonal approaches, we establish that the high glycolytic flux in glucose continuously consumes free Pi. This restricts mitochondrial access to inorganic phosphate (Pi), and prevents mitochondrial activation. Contrastingly, rewired glucose metabolism with enhanced trehalose production and reduced GAPDH (as in *Δubp3* cells) restores Pi. This collectively results in increased mitochondrial Pi and derepression, while restricting mitochondrial Pi transport prevents activation. We therefore suggest that glycolytic flux-dependent intracellular Pi budgeting is a key constraint for mitochondrial repression.

## eLife assessment

This study provides **valuable** insights into the regulation of metabolic flux between glycolysis and respiration in yeast, particularly focusing on the role of inorganic phosphate. The authors propose a novel mechanism involving Ubp3/Ubp10 that potentially mitigates the Crabtree effect, offering substantial, **solid** evidence through a variety of well-designed assays. This study could reshape our understanding of metabolic regulation with broad biological contexts.

## Introduction

Rapidly proliferating cells have substantial metabolic and energy demands in order to increase biomass (*Cai and Tu, 2012*; *Zhu and Thompson, 2019*). This includes a high ATP demand, obtained from cytosolic glycolysis or mitochondrial oxidative phosphorylation (OXPHOS), to fuel multiple reactions (*Nelson et al., 2008*). Interestingly, many rapidly proliferating cells preferentially rely on ATP from glycolysis/fermentation over mitochondrial respiration even in oxygen-replete conditions, and is the well-known Warburg effect (*Vander Heiden et al., 2009*; *Warburg, 1925*). Many such cells repress mitochondrial processes in high glucose, termed glucose-mediated mitochondrial repression or the Crabtree effect (*Crabtree, 1929*; *De Deken, 1966*). This is observed in tumors (*Vander Heiden et al., 2009*), neutrophils (*Xia et al., 2021*), activated macrophages (*Kornberg, 2020*), stem cells

(*Abdel-Haleem et al., 2017*; *Pacini and Borziani, 2014*; *Tsogtbaatar et al., 2020*), and famously *Saccharomyces cerevisiae* (*De Deken, 1966*). Numerous studies have identified signaling programs or regulators of glucose-mediated mitochondrial repression. However, biochemical programs and regulatory processes in biology evolve around key biochemical constraints (*Cornish-Bowden, 2016*). The biochemical constraints for mitochondrial repression remain unresolved (*Diaz-Ruiz et al., 2011*; *Hammad et al., 2016*).

There are two hypotheses on the biochemical principles driving mitochondrial repression. The first proposes direct roles for glycolytic intermediates in driving mitochondrial respiration, by inhibiting specific mitochondrial outputs (*Díaz-Ruiz et al., 2008*; *Rosas Lemus et al., 2018*). The second hypothesizes that a competition between glycolytic and mitochondrial processes for mutually required metabolites/co-factors such as pyruvate, ADP, or inorganic phosphate (Pi) could determine the extent of mitochondrial repression (*Diaz-Ruiz et al., 2011*; *Hammad et al., 2016*; *Koobs, 1972*). These are not all mutually exclusive, and a combination of these factors might dictate mitochondrial repression in high glucose. However, any hierarchies of importance are unclear (*Rodríguez-Enríquez et al., 2001*), and experimental data for the necessary constraints for mitochondrial repression remains incomplete.

One approach to resolve this question has been to identify regulators of metabolic state under high glucose. Post-translational modifications (PTMs) regulated by signaling systems can regulate mitochondrial repression (*Broach, 2012*; *Hitosugi and Chen, 2014*; *Tripodi et al., 2015*). Ubiquitination is a PTM that regulates global proteostasis (*Hershko and Ciechanover, 1998*; *Komander and Rape, 2012*), but the roles of ubiquitination-dependent processes in regulating mitochondrial repression are poorly explored. Ubiquitination itself is determined by the balance between ubiquitination and deubiquitinase (DUB) dependent deubiquitination (*Pickart and Eddins, 2004*). Little is known about the roles of DUBs in regulating metabolic states, making the DUBs interesting candidate regulators of mitochondrial repression.

In this study, using an *S. cerevisiae* DUB knockout library-based screen, we identified the evolutionarily conserved DUB Ubp3 (mammalian Usp10) as required for mitochondrial repression in high glucose. Loss of Ubp3 resulted in mitochondrial activation, along with a reduction in the glycolytic enzymes - phosphofructokinase 1 (Pfk1) and GAPDH (Tdh2 and Tdh3). This consequently reroutes glucose flux and increases trehalose biosynthesis. This metabolic rewiring increases Pi release from trehalose synthesis, and decreases Pi consumption in glycolysis, to cumulatively increase Pi pools. Using *ubp3Δ* cells along with independent analysis of wild-type (WT) cells, and isolated mitochondrial fractions, we establish that glycolytic flux-dependent Pi allocations to mitochondria determines mitochondrial activity. Through these data, we propose how intracellular Pi balance as controlled by glycolytic flux is a key biochemical constraint for mitochondrial repression.

## Results

### A DUB deletion screen identifies Ubp3 as a regulator of glucose-mediated mitochondrial repression

In cells such as *S. cerevisiae*, high glucose represses mitochondrial activity as well as OXPHOS-dependent ATP synthesis (*De Deken, 1966*; *Postma et al., 1989*) (illustrated in *Figure 1A*). Our initial objective was to identify proteostasis regulators of glucose-mediated mitochondrial repression. We generated and used a DUB deletion strain library of *S. cerevisiae* (*Figure 1—figure supplement 1A*), to unbiasedly identify regulators of mitochondrial repression by measuring the fluorescence intensity of a potentiometric dye Mitotracker CMXRos (illustrated in *Figure 1B*). Using this screen, we identified DUB mutants with altered mitochondrial membrane potential (*Figure 1C*, *Figure 1—figure supplement 1A*). Note: WT cells in a respiratory medium (2% ethanol) were used as a control to estimate maximum mitotracker fluorescence intensity (*Figure 1—figure supplement 1B*).

A prominent 'hit' was the evolutionarily conserved DUB Ubp3 (*Figure 1C*), (homologous to mammalian Usp10) (*Figure 1—figure supplement 1D*). Due to its high degree of conservation across eukaryotes (*Figure 1—figure supplement 1D*) as well as putative roles in metabolism or mitochondrial function (*Isasa et al., 2015*; *Nostramo et al., 2016*; *Ossareh-Nazari et al., 2010*), we focused our further attention on this DUB. Cells lacking Ubp3 showed an ~1.5-fold increase in mitotracker fluorescence (*Figure 1C*, *Figure 1—figure supplement 1A and C*). Cells with catalytically inactive Ubp3 (Ubp3$^{C469A}$) showed increased mitochondrial potential comparable to *ubp3Δ* (*Figure 1D*). This

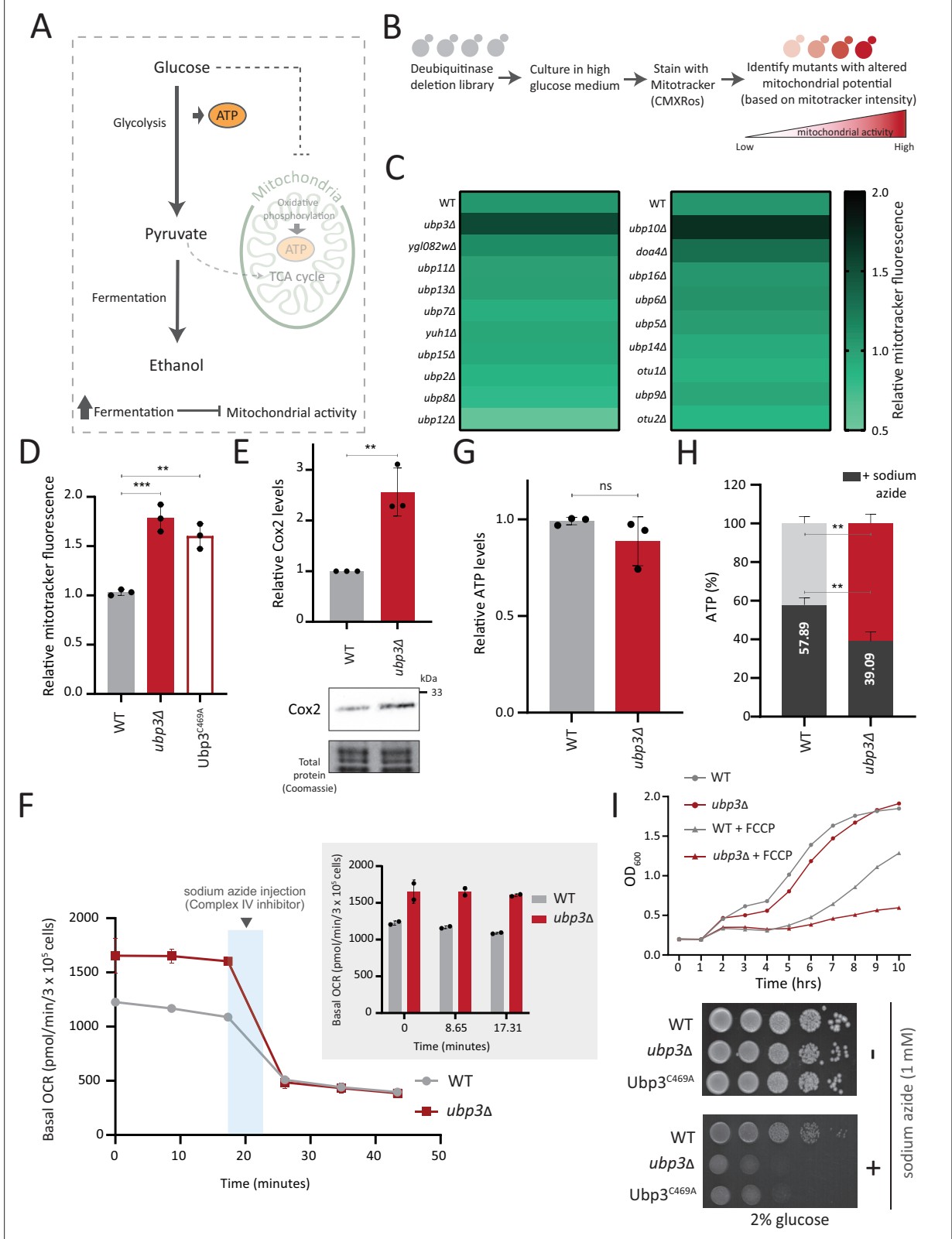

**Figure 1.** A deubiquitinase (DUB) deletion screen identifies Ubp3 as a regulator of glucose-mediated mitochondrial repression. (**A**) Schematic depicting glucose-mediated mitochondrial repression (Crabtree effect). (**B**) Schematic describing the screen with a yeast DUB knockout (KO) library to identify regulators of Crabtree effect. (**C**) Identifying DUB knockouts with altered mitochondrial potential. Heat map shows relative mitochondrial membrane potential of 19 DUB deletions in high glucose, from two biological replicates. Also see *Figure 1—figure supplement 1A and B*. (**D**) The

*Figure 1 continued on next page*

*Figure 1 continued*

DUB activity of Ubp3 and repression of mitochondrial membrane potential. Wild-type (WT), *ubp3Δ*, and Ubp3[C469A] were grown in high glucose and relative mitochondrial membrane potential was measured. Data represent mean ± SD from three biological replicates (n=3). Also see *Figure 1—figure supplement 1D*. (**E**) Effect of loss of Ubp3 on electron transport chain (ETC) complex IV subunit Cox2. WT and *ubp3Δ* were grown in high glucose, and Cox2 was measured (western blot using an anti-Cox2 antibody). A representative blot (out of three biological replicates, n=3) and their quantifications are shown. Data represent mean ± SD. (**F**) Basal oxygen consumption rate (OCR) in high glucose in *ubp3Δ*. WT and *ubp3Δ* were grown in high glucose, and OCR was measured. Basal OCR corresponding to ~3×10^5 cells, from two independent experiments (n=2), normalized to the $OD_{600}$ is shown. Bar graph representations are shown in the inset. Data represent mean ± SD. (**G**) Total ATP levels in *ubp3Δ* and WT. WT and *ubp3Δ* were grown in high glucose, and total ATP were measured. Data represent mean ± SD from three biological replicates (n=3). (**H**) Dependence of *ubp3Δ* on mitochondrial ATP. WT and *ubp3Δ* cells were grown in high glucose, and treated with 1 mM sodium azide for 45 min. Total ATP levels in sodium azide treated and untreated cells were measured. Data represent mean ± SD (n=3). (**I**) Requirement for mitochondrial respiration in high glucose in *ubp3Δ*. A growth curve of WT and *ubp3Δ* in high glucose in the presence of oxidative phosphorylation (OXPHOS) uncoupler FCCP (10 μM), and serial dilution growth assay in high glucose in the presence/absence of sodium azide (1 mM) are shown. Data represent mean ± SD (n=2). Also see *Figure 1—figure supplement 1H and I*. Data information: **p<0.01, ***p<0.001.

The online version of this article includes the following source data and figure supplement(s) for figure 1:

**Source data 1.** Uncropped and labeled gels and blots for *Figure 1*.

**Source data 2.** Raw unedited gels and blots for *Figure 1*.

**Figure supplement 1.** Deubiquitinase (DUB) screen details and further characterization of Ubp3 functions.

**Figure supplement 1—source data 1.** Uncropped and labeled gels and blots for *Figure 1—figure supplement 1*.

**Figure supplement 1—source data 2.** Raw unedited gels and blots for *Figure 1—figure supplement 1*.

data confirmed that Ubp3 catalytic activity is required to fully repress mitochondrial activity under high glucose. The catalytic site mutation did not affect steady-state Ubp3 levels (*Figure 1—figure supplement 1E*).

Next, to assess the requirement of Ubp3 for mitochondrial function, we quantified the electron transport chain (ETC) complex IV subunit Cox2 (*Fontanesi et al., 2006*; *Figure 1E*). *ubp3Δ* had higher Cox2 than WT (*Figure 1E*). As a control, we estimated total mitochondrial content in WT and *ubp3Δ* cells, using either estimates of the structural protein Tom70, or measuring the fluorescence intensity in strains engineered with mitochondrial targeted mNeonGreen (*Dua et al., 2022*). There was no increase in the total mitochondrial volume (estimated by measuring the intensity of mitochondria targeted mNeon green) (*Figure 1—figure supplement 1F*) or mitochondrial outer membrane protein Tom70 (*Figure 1—figure supplement 1G*). This suggests that the increased Cox2 is not merely because of higher total mitochondrial content. We next measured the basal oxygen consumption rate (OCR) of *ubp3Δ*, and basal OCR was higher in *ubp3Δ*, indicating higher respiration (*Figure 1F*).

Next, we asked if mitochondrial ATP synthesis was higher in *ubp3Δ*. The total ATP levels in WT and *ubp3Δ* were comparable (*Figure 1G*). However, upon treatment with the ETC complex IV inhibitor sodium azide, ATP levels in WT were higher than *ubp3Δ*, contributing to ~60% of the total ATP (*Figure 1H*). In contrast, the ATP levels in *ubp3Δ* after sodium azide treatment were ~40% of the total ATP (*Figure 1H*). These data suggest a higher contribution of mitochondrial ATP synthesis toward the total ATP pool in *ubp3Δ*.

We next asked if *ubp3Δ* required higher mitochondrial activity for growth, using a series of mitochondrial activity inhibitors and comparing relative growth. In high glucose, WT cells show minimal growth inhibition in the presence of sodium azide, indicating lower reliance on mitochondrial function (*Figure 1I*). Contrastingly, *ubp3Δ* or Ubp3[C469A] show a severe growth defect in the presence of the mitochondrial ETC complex inhibitors sodium azide and oligomycin, and the mitochondrial OXPHOS uncoupler FCCP (*Figure 1I*, *Figure 1—figure supplement 1H*). Additionally, the loss of Ubp3 in respiration defective cox2-62 cells (*Bonnefoy et al., 2001*) or Rho0 cells (which lacks mitochondrial DNA) resulted in a severe growth defect (*Figure 1—figure supplement 1I*). Deletion of ATP synthase subunits Atp1 and Atp10 also results in a severe growth defect in *ubp3Δ* compared to WT (*Figure 1—figure supplement 1I*). Together, these results indicate that the loss of Ubp3 makes cells dependent on mitochondrial ATP synthesis in high glucose.

In order to address whether the deletion of Ubp3 might increase ubiquitinated proteins and consequent proteostatic stress, we analyzed the global ubquitination state in *ubp3Δ* cells. WT and *ubp3Δ* cells grown under brief (1 hr) heat stress, which increases protein ubiquitination, was used as a control. However, we did not observe a significant increase in the global ubiquitinatin state in *ubp3Δ* cells

(*Figure 1—figure supplement 1J*). This suggests that the altered mitochondrial metabolism in *ubp3Δ* cells is unlikely to be due to general proteostatic stress.

Collectively these data show that in 2% glucose, *ubp3Δ* have high mitochondrial activity, respiration, and rely on this mitochondrial function for ATP production and growth. We therefore decided to use *ubp3Δ* cells to start delineating requirements for glucose-mediated mitochondrial repression.

## Key glycolytic enzymes decrease and glucose flux is rerouted in *ubp3Δ* cells

Glucose-6 phosphate (G6P) is the central node in glucose metabolism where carbon allocations are made toward distinct metabolic arms, primarily: glycolysis, the pentose phosphate pathway (PPP), and trehalose biosynthesis (*Figure 2A*). We first compared amounts of two key ('rate-controlling') glycolytic enzymes - phosphofructokinase 1 (Pfk1), GAPDH isozymes (Tdh2, Tdh3) (*Nelson et al., 2008*), along with the enolase isozymes (Eno1, Eno2) in WT and *ubp3Δ* cells. Pfk1, Tdh2, and Tdh3 substantially decreased in *ubp3Δ* (but not Eno1 and Eno2) (*Figure 2B*, *Figure 2—figure supplement 1A*). Since DUBs can control protein amounts by regulating proteasomal degradation, we asked if the decrease in Pfk1, Tdh2, and Tdh3 in *ubp3Δ* is due to proteasomal degradation. To test this, we measured the levels of these enzymes in *ubp3Δ* after treatment with proteasomal inhibitor MG132. We did not observe any rescue in the levels of these enzymes in MG132-treated samples, suggesting that the decreased levels of these enzymes were not due to increased proteasomal degradation (*Figure 2—figure supplement 1B*). To further understand if these enzyme transcripts are altered in *ubp3Δ*, we measured the mRNA levels of PFK1, TDH2, and TDH3 in WT, *ubp3Δ*, and Ubp3$^{C469A}$ cells. The transcripts of all the three genes in *ubp3Δ* and Ubp3$^{C469A}$ cells decreased (*Figure 2—figure supplement 1C*), suggesting that Ubp3 regulates the transcripts of these glycolytic enzyme genes. The reduction in the Pfk and GAPDH enzyme amounts was intriguing, because the Pfk and GAPDH steps are critical in determining glycolytic flux (*Nelson et al., 2008*; *Shestov et al., 2014*). A reduction in these enzymes could therefore decrease glycolytic flux, and would reroute glucose (G6P) allocations via mass action toward other branches of glucose metabolism, primarily the PPP as well as trehalose biosynthesis (*Figure 2A*). To assess this, we first measured the steady-state levels of key glycolytic and PPP intermediates, and trehalose in WT or *ubp3Δ* using targeted LC-MS/MS (*Figure 2C*, *Figure 2—figure supplement 1D*). Glucose-6/fructose-6 phosphate (G6P/F6P) increased in *ubp3Δ* (*Figure 2C*). Concurrently, trehalose, and the PPP intermediates ribose 5-phosphate (R5P) and sedoheptulose 7-phosphate (S7P), increased in *ubp3Δ* (*Figure 2C*).

Since steady-state metabolite amounts cannot separate production from utilization, in order to unambiguously assess if glycolytic flux is reduced in *ubp3Δ* cells, we utilized a pulse labeling of $^{13}C_6$ glucose, following which the label incorporation into glycolytic and other intermediates was measured. Note that because glycolytic flux is very high in yeast, this experiment would require rapid pulsing and extraction of metabolites in order to stay in a linear range and avoid label saturation. We therefore established a very short time point of label addition, quenching and metabolite extraction post $^{13}C$ glucose pulse. Since flux saturates/reaches steady state in seconds, we first ensured that the label incorporation into individual metabolites after the $^{13}C$ glucose pulse was in the linear range, and for early glycolytic intermediates this was seconds after glucose addition. This new methodology is extensively described in Materials and methods, with required controls shown in *Figure 2—figure supplement 1F*. WT and *ubp3Δ* cells were grown in high glucose, pulsed with $^{13}C_6$ glucose, and the relative $^{13}C$ label incorporation into glycolytic intermediates and trehalose were measured, as shown in the schematic (*Figure 2—figure supplement 1E*). In *ubp3Δ*, $^{13}C$ label incorporation into G6P/F6P as well as trehalose substantially increased (*Figure 2D*). Contrastingly, $^{13}C$ label incorporation into glycolytic intermediates F1,6BP, G3P, 3PG, and PEP decreased, indicating decreased glycolytic flux (*Figure 2D*). We next measured ethanol concentrations and production rates, as an additional output of relative glycolytic rates. We observed decreased steady-state ethanol levels, as well as ethanol production rates in *ubp3Δ* (*Figure 2F*).

Glycolysis-derived pyruvate is transported to mitochondria and fuels the trichloroacetic acid (TCA) cycle. Therefore, we asked if the decreased glycolytic rate result in a decrease in the TCA cycle flux as well. To test this, we first measured the steady-state levels of TCA cycle intermediates in WT or *ubp3Δ* using targeted LC-MS/MS. We did not observe any significant change in the levels of TCA cycle intermediates in *ubp3Δ*, except malate which showed a significant decrease in *ubp3Δ* (*Figure 2—figure*

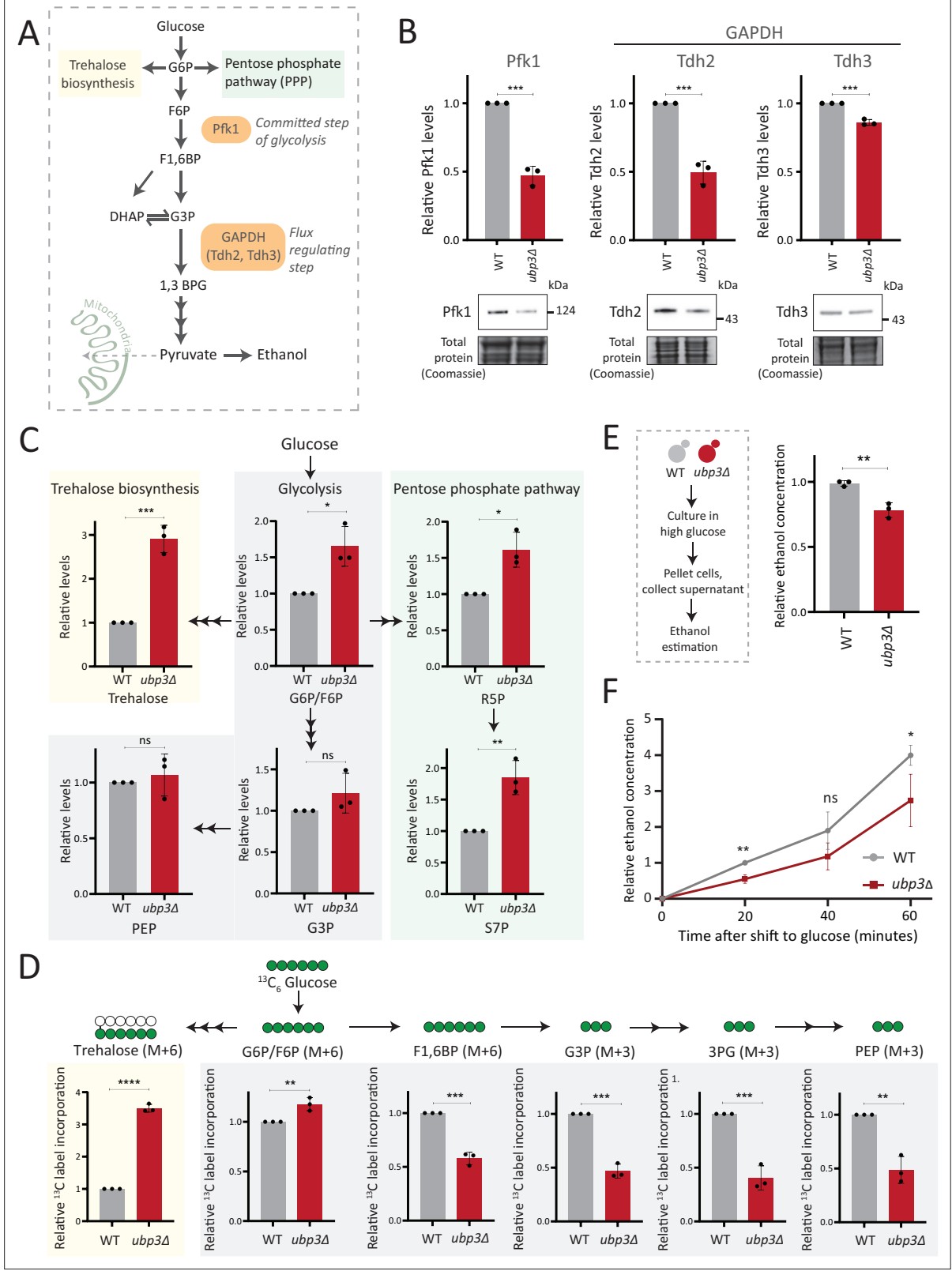

**Figure 2.** Key glycolytic enzymes decrease and glucose flux is rerouted in *ubp3Δ* cells. (**A**) A schematic illustrating directions of glucose-6 phosphate (G6P) flux in cells. Glucose is converted to G6P, a precursor for trehalose, the pentose phosphate pathway (PPP), and glycolysis. (**B**) Effect of loss of Ubp3 on key glycolytic enzymes. Wild-type (WT) and *ubp3Δ* were grown in high glucose and the Pfk1, Tdh2, and Tdh3 levels were measured by western blot using an anti-FLAG antibody. A representative blot (out of three biological replicates, n=3) and their quantification are shown. Data represent mean ±

*Figure 2 continued on next page*

*Figure 2 continued*

SD. Also see *Figure 2—figure supplement 1A*. (**C**) Steady-state metabolite amounts in WT and *ubp3Δ* in high glucose. Relative steady-state levels of trehalose, major glycolytic, and PPP intermediates were estimated in WT and *ubp3Δ*. Data represent mean ± SD from three biological replicates (n=3). Also see *Appendix 1—table 3*. (**D**) Relative glycolytic and trehalose synthesis flux in WT and *ubp3Δ*. Relative $^{13}$C-label incorporation into trehalose and glycolytic intermediates, after a pulse of 1% $^{13}$C$_6$ glucose is shown. Data represent mean ± SD from three biological replicates (n=3). Also see *Appendix 1—table 3*, *Figure 2—figure supplement 1D and E*. (**E**) Ethanol production in *ubp3Δ*. WT and *ubp3Δ* were grown in high glucose and ethanol in the media was measured. Data represent mean ± SD from three biological replicates (n=3). (**F**) Relative rate of ethanol production in WT vs *ubp3Δ*. WT and *ubp3Δ* were grown in high glucose (to OD$_{600}$~0.6), equal numbers of cells were shifted to fresh medium (high glucose) and ethanol concentration in the medium was measured temporally. Data represent mean ± SD from three biological replicates (n=3). Data information: *p<0.05, **p<0.01, ***p<0.001.

The online version of this article includes the following source data and figure supplement(s) for figure 2:

**Source data 1.** Uncropped and labeled gels and blots for *Figure 2*.

**Source data 2.** Raw unedited gels and blots for *Figure 2*.

**Figure supplement 1.** Further estimates of glycolytic enzymes and flux.

**Figure supplement 1—source data 1.** Uncropped and labeled gels and blots for *Figure 2—figure supplement 1*.

**Figure supplement 1—source data 2.** Raw unedited gels and blots for *Figure 2—figure supplement 1*.

**Figure supplement 2.** Estimation of steady-state levels and flux of trichloroacetic acid (TCA) cycle intermediates.

*supplement 2A*). Next, in order to assess if TCA cycle flux reduces in *ubp3Δ* cells, WT and *ubp3Δ* cells were grown in high glucose, pulsed with $^{13}$C$_6$ glucose, and the relative $^{13}$C label incorporation into TCA cycle intermediates was measured, as shown in the schematic (*Figure 2—figure supplement 2B*). The kinetics of $^{13}$C label incorporation in TCA cycle intermediates are shown in *Figure 2—figure supplement 2C*. We did not observe any significant change in the relative $^{13}$C label incorporation in TCA cycle intermediates in *ubp3Δ* (*Figure 2—figure supplement 2D*). Therefore, these data suggest that the decreased glycolytic flux in *ubp3Δ* does not result in a decrease in TCA cycle flux. The increased respiration and mitochondrial activity in *ubp3Δ* cells is therefore driven via other factors.

Collectively, these results reveal that that reduced Pfk1 and GAPDH (in *ubp3Δ*) decrease glucose flux via glycolysis, which results in rewired glucose flux toward trehalose biosynthesis and the PPP.

## Rerouted glucose flux results in phosphate (Pi) accumulation

We therefore asked if the proteomic state, as observed in *ubp3Δ* cells, could provide clues to explain the coupling between mitochondrial derepression and rerouted glucose flux. A recent study by Isasa et al. had systematically quantified the changes in protein levels in *ubp3Δ* cells (*Isasa et al., 2015*). We therefore reanalyzed this extensive dataset, looking for changes in proteins that would correlate with these metabolic processes. Notably, we observed increased levels of proteins of the mitochondrial ETC and respiration, and decreased amounts of glucose metabolizing enzymes in *ubp3Δ* (*Figure 3A*). Additionally, multiple proteins involved in regulating phosphate (Pi) homeostasis were decreased in *ubp3Δ* (*Figure 3A*; *Isasa et al., 2015*). We had recently uncovered a reciprocal coupling of Pi homeostasis with the different arms of glucose metabolism, particularly trehalose biosynthesis (*Gupta et al., 2019*; *Gupta and Laxman, 2021*), and therefore wondered if a glycolytic flux-dependent change in Pi homeostasis had any role in mitochondrial respiration. Our hypothesis was refined based on the reasoning given below.

One explanation for mitochondrial repression can be an internal competition for shared metabolites/co-factors between (cytosolic) glycolytic and mitochondrial processes, that mitochondria might not be sufficiently able to access (*Diaz-Ruiz et al., 2011*; *Koobs, 1972*; *Rodríguez-Enríquez et al., 2001*). In this context, a plausible role for inorganic phosphate (Pi) in regulating mitochondrial repression can be hypothesized. Cytosolic glycolysis requires rapid, high consumption of net Pi (*Mason et al., 1981*; *Rodríguez-Enríquez et al., 2001*; *van Heerden et al., 2014*), and this could possibly limit the Pi that is continuously available for mitochondrial use (*Brazy et al., 1982*; *Koobs, 1972*), thereby repressing mitochondria. Contextually, the balance between reactions releasing vs consuming Pi could explain changes in global Pi levels (*Gupta and Laxman, 2021*). Glycolysis is a continuous hub of Pi consumption. In glycolysis, GAPDH catalyzes G3P to 1,3BPG, converting ADP to ATP, while concurrently consuming a molecule of Pi (*Hohmann et al., 1996*; *van Heerden et al., 2014*). This Pi that goes into ATP will subsequently be used for nucleotide biosynthesis, polyphosphate biosynthesis,

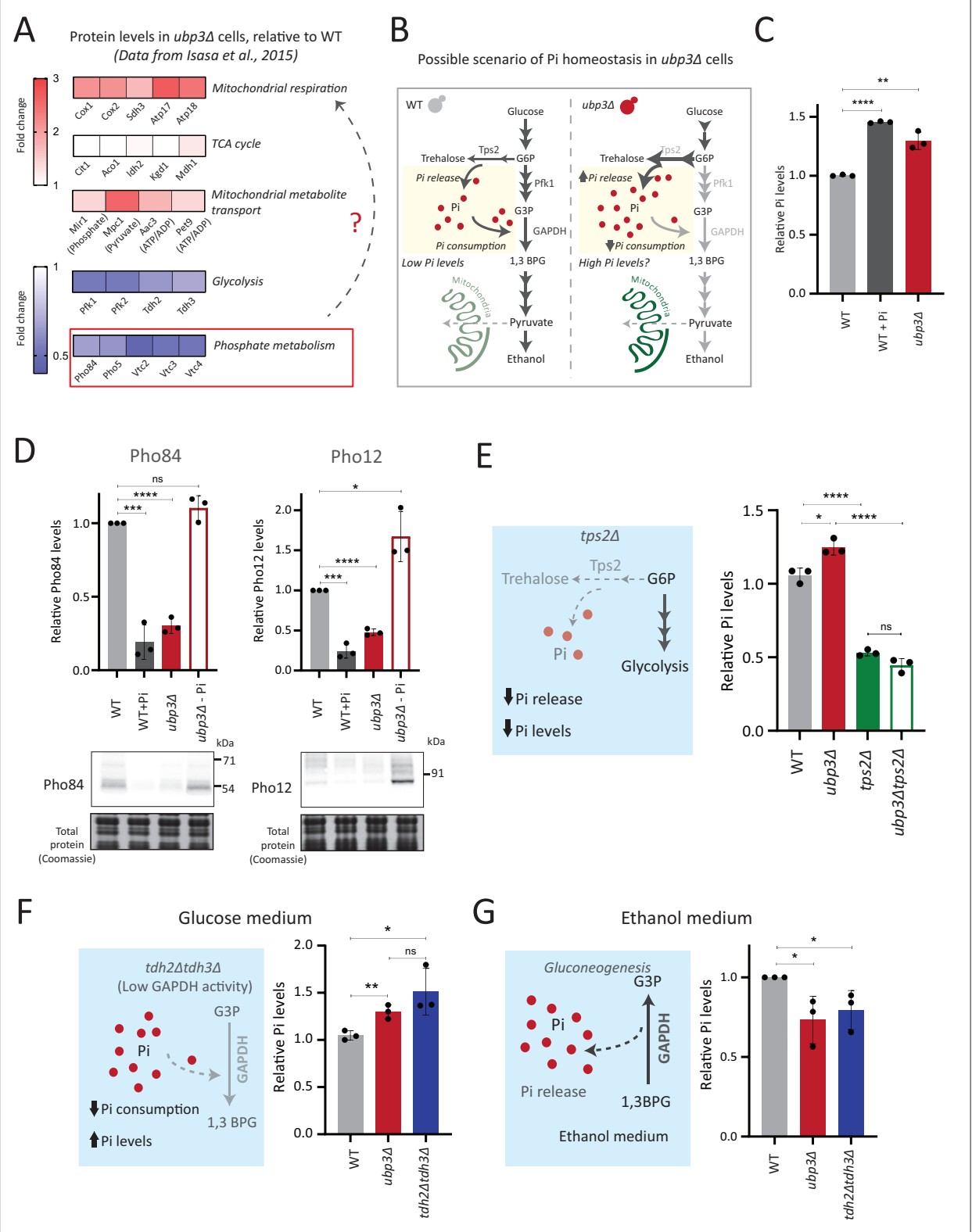

**Figure 3.** Rerouted glucose flux results in inorganic phosphate (Pi) accumulation. (**A**) Changes in protein levels in *ubp3Δ* (dataset from *Isasa et al., 2015*). *ubp3Δ* cells have an increase in proteins involved in mitochondrial respiration and decrease in proteins involved in glucose and phosphate metabolism. (**B**) Schematic showing maintenance of Pi balance during glycolysis. Trehalose synthesis from glucose-6 phosphate (G6P) releases Pi, and the conversion of G3P to 1,3BPG by GAPDH consumes Pi. In *ubp3Δ*, trehalose biosynthesis (which releases Pi) increases. *ubp3Δ* have decreased GAPDH, which will decrease Pi consumption. This increase in Pi release along with decreased Pi consumption could increase cytosolic Pi. (**C**) Intracellular

*Figure 3 continued on next page*

*Figure 3 continued*

Pi levels in wild-type (WT) and *ubp3Δ*. WT and *ubp3Δ* were grown in high glucose and the total free phosphate (Pi) levels were estimated. WT in high Pi (2% glucose, 10 mM Pi) was a positive control. Data represent mean ± SD from three biological replicates (n=3). Also see *Figure 3—figure supplement 1A*. (**D**) Pho regulon responses in WT and *ubp3Δ*. Protein levels of Pho84-FLAG and Pho12-FLAG were compared between WT grown in high glucose and in high Pi, *ubp3Δ* in high glucose with or without a shift to a no-Pi medium for 1 hr, by western blot. A representative blot (out of three biological replicates, n=3) and their quantifications are shown. Data represent mean ± SD. (**E**) Contribution of trehalose synthesis as a Pi source. WT, *tps2Δ*, *ubp3Δ*, and *ubp3Δtps2Δ* were grown in high glucose and the total Pi levels were estimated. Data represent mean ± SD from three biological replicates (n=3). Also see *Figure 3—figure supplement 1B*. (**F**) Loss of GAPDH isozymes Tdh2 and Tdh3 and effect on Pi. WT, *ubp3Δ*, and *tdh2Δtdh3Δ* were grown in high glucose and total Pi was estimated. Data represent mean ± SD from three biological replicates (n=3). (**G**) Pi levels in *ubp3Δ* and *tdh2Δtdh3Δ* cells in ethanol medium. WT, *ubp3Δ*, and *tdh2Δtdh3Δ* cells were grown in ethanol medium and the total Pi levels were estimated. Data represent mean ± SD from three biological replicates (n=3). Data information: *p<0.05, **p<0.01, ***p<0.001, ****p<0.0001.

The online version of this article includes the following source data and figure supplement(s) for figure 3:

**Source data 1.** Uncropped and labeled gels and blots for *Figure 3*.

**Source data 2.** Raw unedited gels and blots for *Figure 3*.

**Figure supplement 1.** Comparisons of phosphate, ethanol, and other metabolites in wild-type (WT), *ubp3Δ* and GAPDH mutants.

and protein phosphorylation (*Gupta and Laxman, 2021*; *Hunter, 2012*). Therefore, we can surmise that in high glycolytic flux, the production of ATP, nucleotides, and polyphosphates is concurrent with Pi consumption (*Austin and Mayer, 2020*; *Hohmann et al., 1996*; *Ljungdahl and Daignan-Fornier, 2012*). Could this reaction therefore limit cytosolic Pi for the mitochondria (as illustrated in *Figure 3B*)? Notably, *ubp3Δ* have reduced GAPDH levels and decreased glycolytic flux. Second, trehalose synthesis is a Pi-releasing reaction, and a major source of free Pi that is critical for Pi homeostasis (*Gupta et al., 2019*; *van Heerden et al., 2014*). Flux through this reaction is also substantially higher in *ubp3Δ*. Therefore, these cells might have increased Pi release (via trehalose), coupled with decreased Pi consumption (via decreased GAPDH). We therefore asked if total Pi increases in *ubp3Δ* (*Figure 3B*)?

To test this, we directly assessed total Pi levels in *ubp3Δ* and WT. *ubp3Δ* cells had higher Pi than WT, and this was comparable to Pi in WT grown in excess Pi (*Figure 3C*). Similarly, Pi amounts also increased in Ubp3$^{C469A}$ (*Figure 3—figure supplement 1A*). Therefore, the loss of Ubp3 increases intracellular Pi levels. Next, we asked if *ubp3Δ* cells exhibit signatures of a 'high Pi' state. *S. cerevisiae* maintains internal Pi balance by controlling the expression of multiple genes collectively known as the Pho regulon (*Mouillon and Persson, 2006*). The Pho regulon is induced under Pi limitation, and repressed during Pi sufficiency (*Gupta et al., 2019*; *Mouillon and Persson, 2006*). We assessed two major Pho proteins (Pho84: a high-affinity membrane Pi transporter, and Pho12: an acid phosphatase) in WT and *ubp3Δ* in high glucose. *ubp3Δ* cells have lower amounts of Pho84 and Pho12 (*Figure 3D*). Further, upon shifting to low Pi for 1 hr, Pho84 and Pho12 increased in *ubp3Δ*. These data suggest that reduced Pho84 and Pho12 amounts in *ubp3Δ* are because of increased Pi, and not due to altered Pho regulon function itself (*Figure 3D*). These data clarify earlier observations from *ubp3Δ* which noted reduced Pho proteins (*Isasa et al., 2015*). Therefore, *ubp3Δ* constitutively have higher Pi, and likely a consequent decrease in Pho proteins.

We next asked if the increased Pi in *ubp3Δ* is because of altered G6P allocations toward different end-points, particularly trehalose synthesis, which can be a major node of Pi restoration (*Gupta et al., 2019*; *van Heerden et al., 2014*). We assessed the contribution of increased trehalose synthesis toward the high Pi in *ubp3Δ*, by estimating Pi levels in the absence of trehalose 6-phosphate phosphatase (Tps2), which catalyzes the Pi-releasing step in trehalose synthesis. Notably, loss of Tps2 in *ubp3Δ* decreased Pi (*Figure 3E*). There was no additive difference in Pi between *tps2Δ* and *ubp3Δtps2Δ* (*Figure 3E*). As an added control, we assessed trehalose in WT and *ubp3Δ* in the absence of Tps2, and found no difference (*Figure 3—figure supplement 1B*). Therefore, increasing G6P flux toward trehalose biosynthesis is a major source of the increased Pi in *ubp3Δ*.

Since the major GAPDH isozymes, Tdh2 and Tdh3, are reduced in *ubp3Δ*, we directly asked if reducing GAPDH can decrease Pi consumption and increase Pi. To assess this, we generated *tdh2Δtdh3Δ* cells, which exhibit a growth defect, but are viable, permitting further analysis. *tdh2Δtdh3Δ* had higher Pi in high glucose (*Figure 3F*). Expectedly, we observed a decrease in ethanol in *tdh2Δtdh3Δ* (*Figure 3—figure supplement 1C*), along with an accumulation of F1,6BP, and G3P, and decreased 3PG and PEP (*Figure 3—figure supplement 1D*). However, G6P and trehalose levels

between WT and *tdh2Δtdh3Δ* were comparable (*Figure 3—figure supplement 1D and E*). These data suggest that unlike in *ubp3Δ*, the increased Pi in *tdh2Δtdh3Δ* comes mainly from decreased Pi consumption (GAPDH step). To further assess the role of reduced glycolytic flux in increasing Pi (*ubp3Δ* and *tdh2Δtdh3Δ*), we measured the Pi in these cells growing in a gluconeogenic medium - 2%, ethanol. In this scenario, the GAPDH catalyzed reaction will be reversed, converting 1,3BPG to G3P, which should release and not consume Pi. Compared to WT, the Pi levels decreased in *ubp3Δ* and *tdh2Δtdh3Δ* (*Figure 3G*), suggesting that the changes in Pi in these mutants are driven by the relative change in Pi release vs consumption.

These results collectively indicate that the combined effect of increased Pi release coming from trehalose synthesis and decreased Pi consumption from reduced GAPDH increase Pi levels in *ubp3Δ*.

## Mitochondrial Pi availability correlates with mitochondrial activity in *ubp3Δ*

We therefore wondered if this observed Pi increase from the combined rewiring of glucose metabolism resulted in more Pi becoming accessible to the mitochondria. This would effectively result in Pi budgeting between cytosolic glycolysis and mitochondria based on the relative flux in different arms of glucose metabolism, by increasing Pi availability for the mitochondria (*Figure 4A*). If this were indeed so, a prediction would be that the pool of Pi in the mitochondria of *ubp3Δ* cells would be higher than WT cells, while the opposite would be expected in the cytosolic fraction of these cells, due to increased Pi in the mitochondria. To test if this were so, we first measured the cytosolic pools of Pi in WT and *ubp3Δ*. The cytosolic fraction was isolated (extensive experimental details are in the Appendix 1) from WT and *ubp3Δ* cells, and Pi levels in this fraction were estimated (*Figure 4B*, *Figure 4—figure supplement 1A*). We observed significantly reduced Pi in the cytosolic fraction in *ubp3Δ* cells. Since total cellular Pi amounts is higher in *ubp3Δ* cells (*Figure 3*), a decreased cytosolic Pi would be consistent with greater transport of Pi from cytosol to other organelles such as vacuole (where it is stored as polyphosphate) and/or mitochondria. Therefore, we next asked if the mitochondria in *ubp3Δ* have correspondingly increased Pi. Mitochondria were isolated by immunoprecipitation from WT and *ubp3Δ*, the isolation efficiency was analyzed (*Figure 4—figure supplement 1A*), and relative Pi levels compared. Pi levels were normalized to Idh1 (isocitrate dehydrogenase) in isolated mitochondria, since Idh1 protein levels did not decrease in *ubp3Δ* (*Figure 4—figure supplement 1B*), and consistent with another study (*Isasa et al., 2015*). Mitochondrial Pi was higher in *ubp3Δ* (*Figure 4B*). We next asked if the increased mitochondrial activity in *ubp3Δ* is a consequence of high Pi. If this were so, *tdh2Δtdh3Δ* should partially phenocopy *ubp3Δ* with respect to mitochondrial activity. Consistently, *tdh2Δtdh3Δ* have higher mitotracker intensity as well as Cox2 levels compared to WT (*Figure 4—figure supplement 1C*, *Figure 4C*). Also consistent with this, a high basal OCR in *tdh2Δtdh3Δ* was observed (*Figure 4D*), together indicating high mitochondrial activity in *tdh2Δtdh3Δ*, which is comparable to *ubp3Δ*.

We next asked if higher Pi is necessary to increase mitochondrial activity in *ubp3Δ* and *tdh2Δtdh3Δ*. Since glycolysis is defective in *ubp3Δ* and *tdh2Δtdh3Δ*, it is necessary to distinguish the effect of high Pi vs. only the effect of low glycolysis in activating mitochondria. Logically, if decreased glycolysis (independent of Pi) is sufficient to activate mitochondria, bringing down the Pi levels in *ubp3Δ* to that of WT should not affect mitochondrial activity. Notably, *ubp3Δ* grown in low (1 mM) Pi have Pi levels similar to WT in standard (normal Pi) medium (*Figure 4—figure supplement 1D*). *ubp3Δ* grown in low Pi also had decreased ethanol, suggesting reduced glycolysis (*Figure 4—figure supplement 1E*). Therefore, we used this condition to further understand the role of Pi in inducing mitochondrial activity. Mitotracker fluorescence decreased in both *ubp3Δ* and *tdh2Δtdh3Δ* in low Pi (*Figure 4—figure supplement 1F*). Note: Basal mitotracker fluorescence in WT also decreases in low Pi, which is consistent with a required role of Pi for mitochondrial activity (*Figure 4—figure supplement 1F*). Similarly, Cox2 levels were reduced in both *ubp3Δ* and *tdh2Δtdh3Δ* (*Figure 4E*). Consistent with both reduced mitotracker intensity and Cox2 levels, basal OCR also decreased in both *ubp3Δ* and *tdh2Δtdh3Δ* in low Pi (*Figure 4F*). As an additional control, we used Rho0 strains (which have no mitochondrial DNA and therefore lack functional ETC) to compare basal OCR. The expectation in these strains is that basal OCR will not change if Pi changes. Consistently, we did not observe any significant difference in basal OCR in WT, *ubp3Δ* and *ubp3Δ* in low Pi in a Rho0 strain background (which lacks mitochondrial DNA) (*Figure 4—figure supplement 1G*). These data

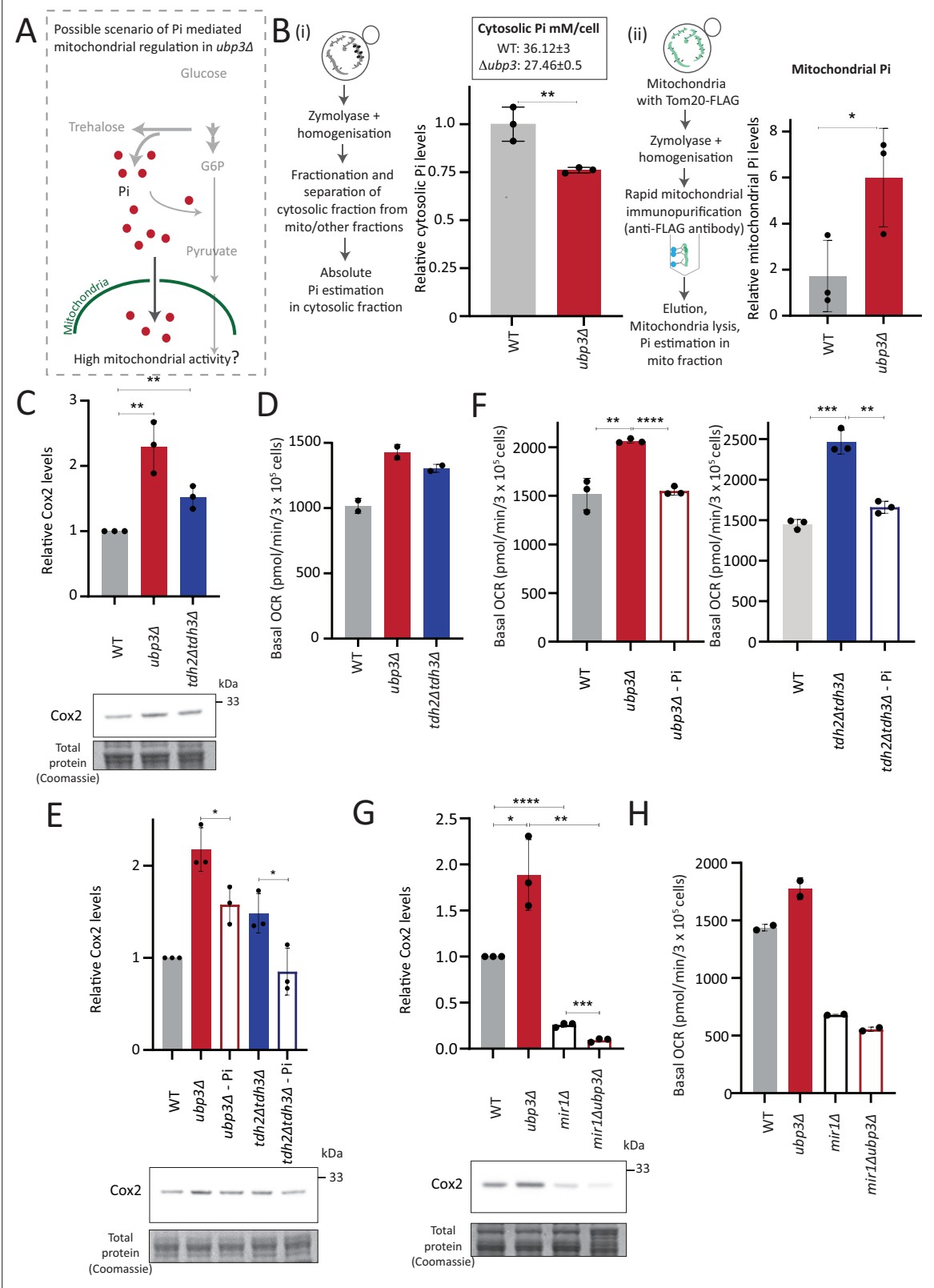

**Figure 4.** Mitochondrial inorganic phosphate (Pi) availability correlates with mitochondrial activity in *ubp3Δ*. (**A**) A hypothetical mechanism of cytosolic free Pi controlling mitochondrial activity by regulating mitochondrial Pi availability. (**B**) Cytosolic and mitochondrial Pi amounts in wild-type (WT) vs *ubp3Δ*. The cytosolic fraction was isolated by centrifugation (see Appendix 1), and in separate experiments, mitochondria were isolated by immunoprecipitation from WT and *ubp3Δ* and mitochondrial Pi estimated. (**i**) Cytosolic Pi levels (relative as well as absolute) and (**ii**) mitochondrial Pi

*Figure 4 continued on next page*

*Figure 4 continued*

levels (normalized to Idh1) are shown. Data represent mean ± SD from three biological replicates (n=3) respectively for the cytosolic and mitochondrial measurements. Also see *Figure 4—figure supplement 1A and B*. (**C**) Cox2 protein in *tdh2Δtdh3Δ*. WT, *ubp3Δ*, and *tdh2Δtdh3Δ* were grown in high glucose and Cox2 protein was estimated. A representative blot (out of three biological replicates, n=3) and their quantifications are shown. Data represent mean ± SD. (**D**) Basal oxygen consumption rate (OCR) levels in *tdh2Δtdh3Δ*. WT, *ubp3Δ*, and *tdh2Δtdh3Δ* were grown in high glucose and basal OCR was measured from two independent experiments (n=2). Data represent mean ± SD. Also see *Figure 4—figure supplement 1C*. (**E**) Comparative Pi amounts and Cox2 levels in *ubp3Δ*, *tdh2Δtdh3Δ*, WT cells. WT cells were grown in high glucose, *ubp3Δ* and *tdh2Δtdh3Δ* were grown in high glucose and low Pi, and Cox2 protein was estimated. A representative blot (out of three biological replicates, n=3) and their quantifications are shown. Data represent mean ± SD. Also see *Figure 4—figure supplement 1D and F*. (**F**) Pi amounts and basal OCR in *ubp3Δ* and *tdh2Δtdh3Δ* vs WT cells. WT cells were grown in high glucose, *ubp3Δ* and *tdh2Δtdh3Δ* were grown in high glucose and low Pi, and basal OCR was measured from three independent experiments (n=3). Data represent mean ± SD. (**G**) Effect of loss of mitochondrial Pi transporter Mir1 on Cox2 protein. WT, *ubp3Δ*, *mir1Δ*, and *mir1Δubp3Δ* were grown in high glucose and Cox2 amounts compared. A representative blot (out of three biological replicates, n=3) and their quantifications are shown. Data represent mean ± SD. (**H**) Relationship of mitochondrial Pi transport and basal OCR in WT vs *ubp3Δ*. WT, *ubp3Δ*, *mir1Δ*, and *mir1Δubp3Δ* cells were grown in high glucose and basal OCR was measured from two independent experiments (n=2). Data represent mean ± SD. Data information: *p<0.05, **p<0.01, ****p<0.0001.

The online version of this article includes the following source data and figure supplement(s) for figure 4:

**Source data 1.** Uncropped and labeled gels and blots for *Figure 4*.

**Source data 2.** Raw unedited gels and blots for *Figure 4*.

**Figure supplement 1.** Mitochondrial inorganic phosphate (Pi) estimation characterizations and correlations of mitochondrial activity with Pi availability.

**Figure supplement 1—source data 1.** Uncropped and labeled gels and blots for *Figure 4—figure supplement 1*.

**Figure supplement 1—source data 2.** Raw unedited gels and blots for *Figure 4—figure supplement 1*.

collectively suggest that high intracellular Pi is necessary to increase mitochondrial activity in *ubp3Δ* and *tdh2Δtdh3Δ*.

Next, we asked how mitochondrial Pi transport regulates mitochondrial activity. Mir1 and Pic2 are mitochondrial Pi transporters, with Mir1 being the major Pi transporter (*Murakami et al., 1990*; *Zara et al., 1996*). We first limited mitochondrial Pi availability in *ubp3Δ* by knocking out *MIR1*. In *mir1Δ*, the increased Cox2 observed in *ubp3Δ* was no longer observed (*Figure 4G*). Consistent with this, we observed no further increase in the basal OCR in *mir1Δubp3Δ* compared to *mir1Δ* (*Figure 4H*). Furthermore, *mir1Δ* showed decreased Cox2 as well as basal OCR even in WT cells (*Figure 4G and H*). These data together suggest that mitochondrial Pi transport is critical for increasing mitochondrial activity in *ubp3Δ*, and in maintaining basal mitochondrial activity even in high glucose.

As a control, no significant increase in Mir1 and Pic2 was observed in *ubp3Δ* (*Figure 4—figure supplement 1H*), suggesting that *ubp3Δ* do not increase mitochondrial Pi by merely increasing the Pi transporters, but rather by increasing available Pi pools.

Taken together, these data suggest the possibility that the altered Pi homeostasis in *ubp3Δ* cells increases the mitochondrial Pi pool. This increased mitochondrial Pi pool correlates with increased mitochondrial activity. Decreasing mitochondrial Pi by either reducing total Pi or by reducing mitochondrial Pi transport decreases mitochondrial activity.

## Mitochondrial Pi availability constrains mitochondrial activity under high glucose

So far, these data suggest that the cytosolic Pi available for the mitochondria can determine the extent of mitochondrial activity. Therefore, we further investigated if mitochondrial Pi allocation was a necessary constraint for glucose-mediated mitochondrial repression.

To test this, we first asked how important mitochondrial Pi transport was to switch to increased respiration. In WT yeast, low glucose or glycolytic inhibition will result in increased respiration (*Broach, 2012*). What happens therefore if we restrict mitochondrial Pi in this context? For this, we measured the basal OCR in WT and *mir1Δ* after switching from high (2%) to low (0.1%) glucose. We observed a significant increase in the basal OCR in WT but not in *mir1Δ* (*Figure 5A*). The alternate scenario is after glycolytic inhibition. We assessed the role of mitochondrial Pi in this context, by inhibiting glycolysis using 2-deoxyglucose (2DG). WT, but not *mir1Δ*, increased their OCR (respiration) upon a 1 hr treatment with 2DG (*Figure 5B*). Consistent with this, mitotracker fluorescence increased with an increase in 2DG in WT, but not in *mir1Δ* (*Figure 5—figure supplement 1A*). We further asked if the mitochondrial Pi transporter itself glucose repressed, and therefore assessed Mir1 amounts in high

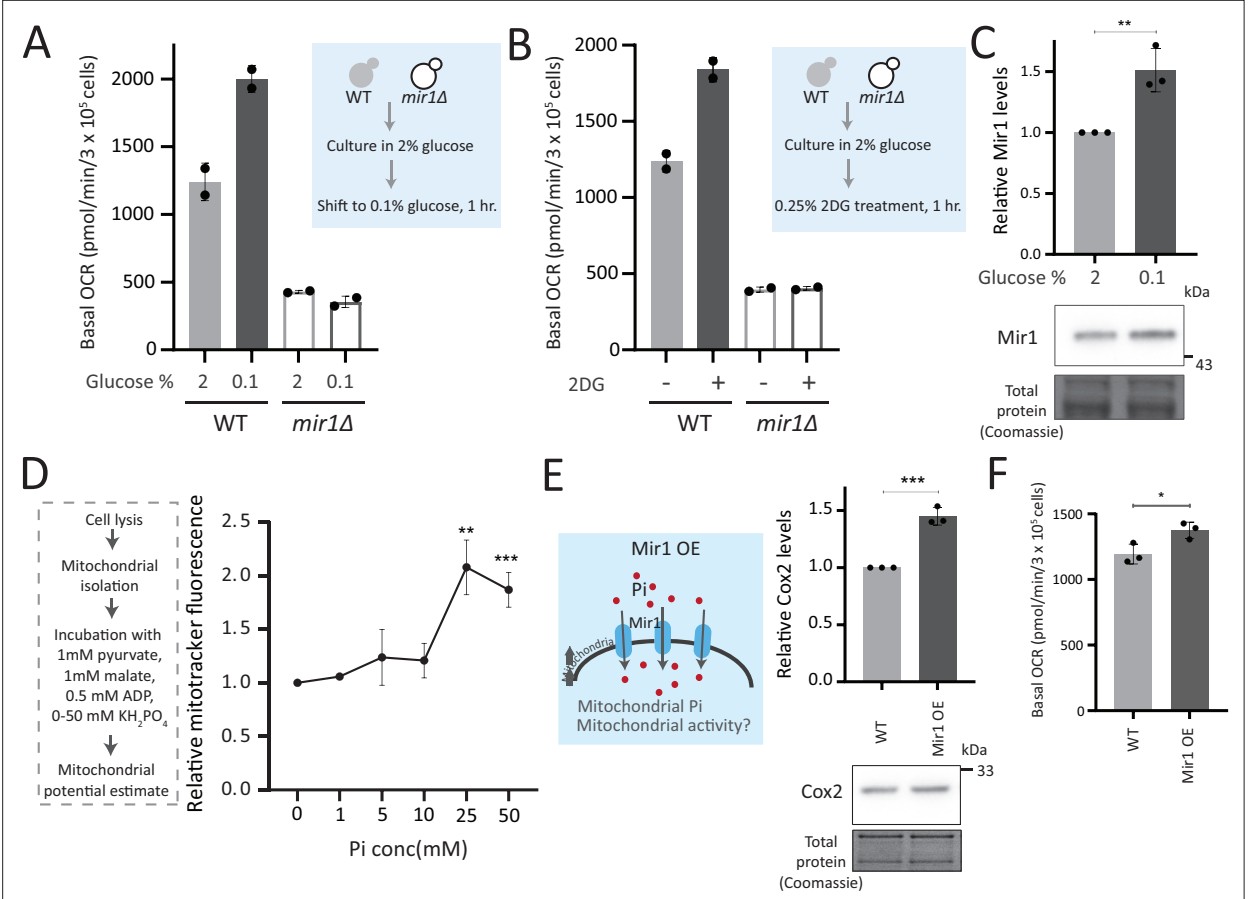

**Figure 5.** Mitochondrial inorganic phosphate (Pi) availability constrains mitochondrial activity under high glucose. (**A**) Relationship of mitochondrial Pi transport and respiration after glucose removal. Wild-type (WT) and *mir1Δ* cells were cultured in high (2%) glucose and shifted to low (0.1%) glucose for 1 hr. The normalized basal oxygen consumption rate (OCR) from two independent experiments (n=2) are shown. Data represent mean ± SD. (**B**) Requirement of mitochondrial Pi transport for switch to respiration upon glycolytic inhibition by 2-deoxyglucose (2DG). WT and *mir1Δ* cells were cultured in high glucose and treated with or without 0.25% 2DG for 1 hr. Basal OCR was measured from two independent experiments (n=2). Data represent mean ± SD. Also see *Figure 5—figure supplement 1A*. (**C**) Glucose-dependent regulation of Mir1. Cells (with Mir1-HA) were grown in high glucose and shifted to low glucose (0.1% glucose) for 1 hr, and Mir1 levels compared. A representative blot (out of three biological replicates, n=3) and their quantifications are shown. Data represent mean ± SD. Also see *Figure 5—figure supplement 1B*. (**D**) Increasing Pi concentrations and mitochondrial activity in isolated mitochondria. Mitochondria were isolated from WT cells grown in high glucose, incubated with 1 mM pyruvate, 1 mM malate, 0.5 mM ADP, and 0–50 mM $KH_2PO_4$. The mitochondrial activity was estimated by mitotracker fluorescence intensity, and intensities relative to the sample with 0 mM $KH_2PO_4$ is shown. Data represent mean ± SD from three biological replicates (n=3). (**E**) Effect of overexpressing Mir1 on Cox2 protein. WT (containing empty vector) and Mir1 overexpressing (Mir1OE) cells were grown in high glucose and Cox2 levels were estimated. A representative blot (out of three biological replicates, n=3) and their quantifications are shown. Data represent mean ± SD. Also see *Figure 5—figure supplement 1G*. (**F**) Effect of overexpressing Mir1 on basal OCR. The basal OCR in WT (containing empty vector) and Mir1OE in high glucose was measured from three independent experiments (n=3). Data represent mean ± SD. Data information: *p<0.05, **p<0.01, ***p<0.001.

The online version of this article includes the following source data and figure supplement(s) for figure 5:

**Source data 1.** Uncropped and labeled gels and blots for *Figure 5*.

**Source data 2.** Raw unedited gels and blots for *Figure 5*.

**Figure supplement 1.** Mitochondrial phosphate and pyruvate transport relationships with mitochondrial activity.

**Figure supplement 1—source data 1.** Uncropped and labeled gels and blots for *Figure 5—figure supplement 1*.

**Figure supplement 1—source data 2.** Raw unedited gels and blots for *Figure 5—figure supplement 1*.

and low glucose. Mir1 levels are higher upon a shift to low glucose, and in cells grown in 2% ethanol, suggesting that Mir1 is glucose repressed (*Figure 5C*, *Figure 5—figure supplement 1B*). These data suggest that mitochondrial Pi transport is necessary for increasing mitochondrial activity after glucose derepression.

We next asked whether just adding external Pi was sufficient to increase mitochondrial activity, when cells are in high glucose. In medium supplemented with excess Pi, the internal Pi increases as seen earlier (*Figure 3C*). Therefore, a simplistic assumption would be that the addition of external Pi to cells in high glucose would also increase mitochondrial Pi. However, an alternate possibility presents itself wherein since glycolytic flux is already high in glucose, supplementing Pi will continue to fuel glycolysis. Indeed, this was originally observed by Harden and Young in 1908, where adding Pi increased fermentation (*Harden and Young, 1906*). In such a scenario, there could be an increase in the cytosolic Pi but not the mitochondrial Pi. We estimated the cytosolic and mitochondrial Pi in this condition where excess Pi was externally supplemented. Notably, cells grown in high Pi had increased cytosolic Pi, but decreased mitochondrial Pi (*Figure 5—figure supplement 1C*), without any changes in total mitochondria volume or amounts (*Figure 5—figure supplement 1D*). Furthermore, directly adding Pi to cells growing in high glucose also decreased basal OCR (*Figure 5—figure supplement 1E*), consistent with decreased mitochondrial Pi. These data indicate that in high glucose, simply supplementing Pi will not increase Pi access to the mitochondria, and instead results in an accumulation of Pi in the cytosol (*Figure 5—figure supplement 1C*). We therefore now asked, if we inhibit glycolytic flux and then supplement Pi, what would happen to mitochondrial activity. For this, we treated cells with 2DG, and subsequently added Pi and measured the OCR (*Figure 5—figure supplement 1F*). In this case, supplementing Pi increased the basal OCR (*Figure 5—figure supplement 1F*). Collectively, these data suggest that a combination of decreasing glycolysis and increasing Pi can together increase respiration. Next, in order to directly test mitochondrial activation based on external Pi availability, we isolated mitochondria, and estimated activity in vitro upon adding increasing Pi. Mitochondrial activity increased with increased Pi, with maximum activity observed with 25 mM Pi supplemented (*Figure 5D*). In a complementary experiment, we overexpressed the Mir1 transporter in WT cells, to increase Pi within mitochondria (*Figure 5—figure supplement 1G*). Mir1-OE cells have higher Cox2 levels and basal OCR (*Figure 5E and F*). Therefore, increasing Pi transport to mitochondria is sufficient to increase mitochondrial activity in high glucose.

Mitochondrial pyruvate transport is also required for mitochondrial respiration (*Timón-Gómez et al., 2013*). We asked where Pi availability stands in a hierarchy of constraints for mitochondrial derepression, as compared to mitochondrial pyruvate transport. We measured the amounts of the Mpc3 subunit of the mitochondrial pyruvate carrier (MPC) complex (*Bender et al., 2015*; *Timón-Gómez et al., 2013*). Mpc3 protein increases in *ubp3Δ* (*Figure 5—figure supplement 1H*). This also correlated with the unimpaired TCA cycle flux (*Figure 2—figure supplement 2D*) and the increased mitochondrial activity. Interestingly, in *ubp3Δ* grown in low Pi, Mpc3 further increased (*Figure 5—figure supplement 1H*), but as shown earlier this condition cannot increase OCR or mitochondrial activity (*Figure 4F*, *Figure 4—figure supplement 1F*). Basal Mpc3 levels decrease in *mir1Δ*, but upon shifting to 0.1% glucose, Mpc3 increases in both WT and *mir1Δ*, with higher levels in *mir1Δ* (*Figure 5—figure supplement 1I*). Therefore, even where Mpc3 is high (*ubp3Δ* in low Pi, and *mir1Δ* in low glucose), mitochondrial activity remains low if Pi is restricted (*Figure 4F and H*). There was also no decrease in basal OCR in *mpc3Δ* in high glucose, and the basal OCR increased to the same level as of WT after shifting to 0.1% glucose (*Figure 5—figure supplement 1J*). Since Mpc3 changes with mitochondrial Pi availability (*Figure 5—figure supplement 1H*), we also measured Mpc3 in the Mir1OE. No further changes in Mpc3 were observed in Mir1OE (*Figure 5—figure supplement 1K*), indicating that increasing mitochondrial Pi alone need not increase Mpc3. Overall, although Mpc3 levels correlate with decreased glycolysis (*ubp3Δ* - *Figure 5—figure supplement 1H*, *ubp3Δ* in low Pi - *Figure 5—figure supplement 1H*, low glucose - *Figure 5—figure supplement 1I*), increased Mpc3 alone cannot increase mitochondrial activity and respiration in the absence of adequate mitochondrial Pi.

Collectively, mitochondrial Pi availability constrains glucose-mediated mitochondrial repression. Increasing available pools of Pi to enter the mitochondria is sufficient to induce mitochondrial activity.

## Repression of mitochondrial respiration via Pi budgeting is conserved in Ubp3 mutants across diverse yeast genetic backgrounds

So far, we have identified a role for intracellular Pi budgeting as a constraint for mitochondrial activity under high glucose. These were all carried out using a robust, prototrophic yeast strain from a CEN.PK background. *S. cerevisiae* however, while Crabtree positive, have tremendous genetic diversity (*Peter*

*et al., 2018*). We therefore asked if this mitochondrial repression through Pi budgeting (mediated by Ubp3 function) is conserved across other strains of *S. cerevsiae* as well. To test this, we generated Ubp3 deletion mutants in different genetic backgrounds of *S. cerevsiae*, including BY4742, W303, and Σ1278. In all these strain backgrounds, we observed a significant increase in mitotracker fluorescence intensity and Cox2 protein levels in *ubp3Δ* (*Figure 6A and B*). This suggests the role of Ubp3 as a regulator of mitochondrial repression, independent of the genetic background of the yeast (*S. cerevsiae*) strain. To further assess if loss of Ubp3 shows a concurrent increase in Pi levels, we measured the total Pi levels in WT and *ubp3Δ* in a W303 strain background. We observed a significant increase in total Pi levels in *ubp3Δ* cells in this strain (*Figure 6C*), similar to what we observed in the CEN.PK strain (*Figure 3C*). Finally, to test if the altered Pi budgeting regulates the mitochondrial activity in these cells, we measured the basal OCR in WT, *ubp3Δ*, and *ubp3Δ* in low (1 mM) Pi medium, in the W303 strain background. Consistent with the increase in mitotracker fluorescence intensity and Cox2 protein levels, we observed a significant increase in the basal OCR in *ubp3Δ* cells (*Figure 6D*). This increase was not observed in *ubp3Δ* in a low Pi medium (*Figure 6D*). This suggests that the role of altered Pi budgeting in regulating mitochondrial respiration is conserved in other genetic backgrounds of *S. cerevisiae*.

Finally, we asked how important mitochondrial Pi transport was for growth, under glycolytic inhibition. Consistent with the requirement for mitochondrial Pi transport to increase mitochondrial respiration upon glycolytic inhibition (*Figure 5B*), WT cells only exhibit a slightly decreased growth in the presence of 2DG (*Figure 6E*). In contrast, *mir1Δ* show a severe growth defect upon 2DG treatment (*Figure 6E*), revealing a synergetic effect of combining 2DG with inhibiting mitochondrial Pi transport. Therefore, the combined inhibition of glycolysis and mitochondrial Pi transport restricts the growth of glycolytic cells.

Collectively, our data suggests a conserved role for intracellular Pi budgeting in regulating mitochondrial repression in high glucose and the role of mitochondrial Pi transport in regulating adaptation for growth under glycolytic inhibition.

## Discussion

In this study, we highlight a role for Pi budgeting between cytosolic glycolysis and mitochondrial processes (which compete for Pi) in constraining mitochondrial repression (*Figure 6F*). Ubp3 controls this process (*Figure 1*), by maintaining the amounts of the glycolytic enzymes Pfk1 and GAPDH (Tdh2 and Tdh3) and thereby allows high glycolytic flux. At high fermentation rates, glycolytic enzymes levels at maximal activity maintain high glycolytic rates (effectively following zero-order kinetics), and therefore, changes in the enzyme levels will have a direct, proportionate effect on flux (*Grigaitis and Teusink, 2022*). The loss of Ubp3 decreases glycolytic flux, resulting in a systems-level, mass-action-based rewiring of glucose metabolism where more G6P is routed toward trehalose synthesis and PPP (*Figure 2*, *Figure 6F*). Indeed, inhibiting just phosphofructokinase can reroute glucose flux from glycolysis to PPP (*Hollinshead et al., 2016*; *Miyazawa et al., 2017*; *Yi et al., 2012*), and our study now permits contextualized interpretations of these results. Such reallocations of glucose flux will collectively increase overall Pi, coming from the combined effect of increased Pi release from trehalose synthesis, and decreased Pi consumption via reduced GAPDH. This altered intracellular Pi economy increases Pi pools available to mitochondria, and increasing mitochondrial Pi is necessary and sufficient to increase respiration in high glucose (*Figure 4*, *Figure 5*, *Figure 6*). Mitochondrial Pi transport maintains mitochondrial activity in high glucose, and increases mitochondrial activity in low glucose (*Figure 4*, *Figure 5*). Finally, the mitochondrial Pi transporter Mir1 itself decreases in high glucose (*Figure 5*). Therefore, this glucose-dependent repression of Mir1 also restricts mitochondrial Pi availability (and thereby activity) in high glucose.

Traditionally, loss-of-function mutants of metabolic enzymes are used to understand metabolic state regulation. This approach negates nuanced investigations, since metabolic enzymes are often essential for viability. Furthermore, metabolic pathways have multiple, contextually regulated nodes, through which cells maintain their metabolic state. Therefore, alternate approaches to identify global regulators of metabolic states (as opposed to single enzymes) might uncover ways via which multiple nodes are simultaneously tuned, and can reveal unanticipated systems-level principles of metabolic state rewiring. In this study of glucose-mediated mitochondrial repression, the loss of the DUB Ubp3 decreases glycolytic flux by reducing the enzymes at two critical nodes in the pathway - Pfk1 and

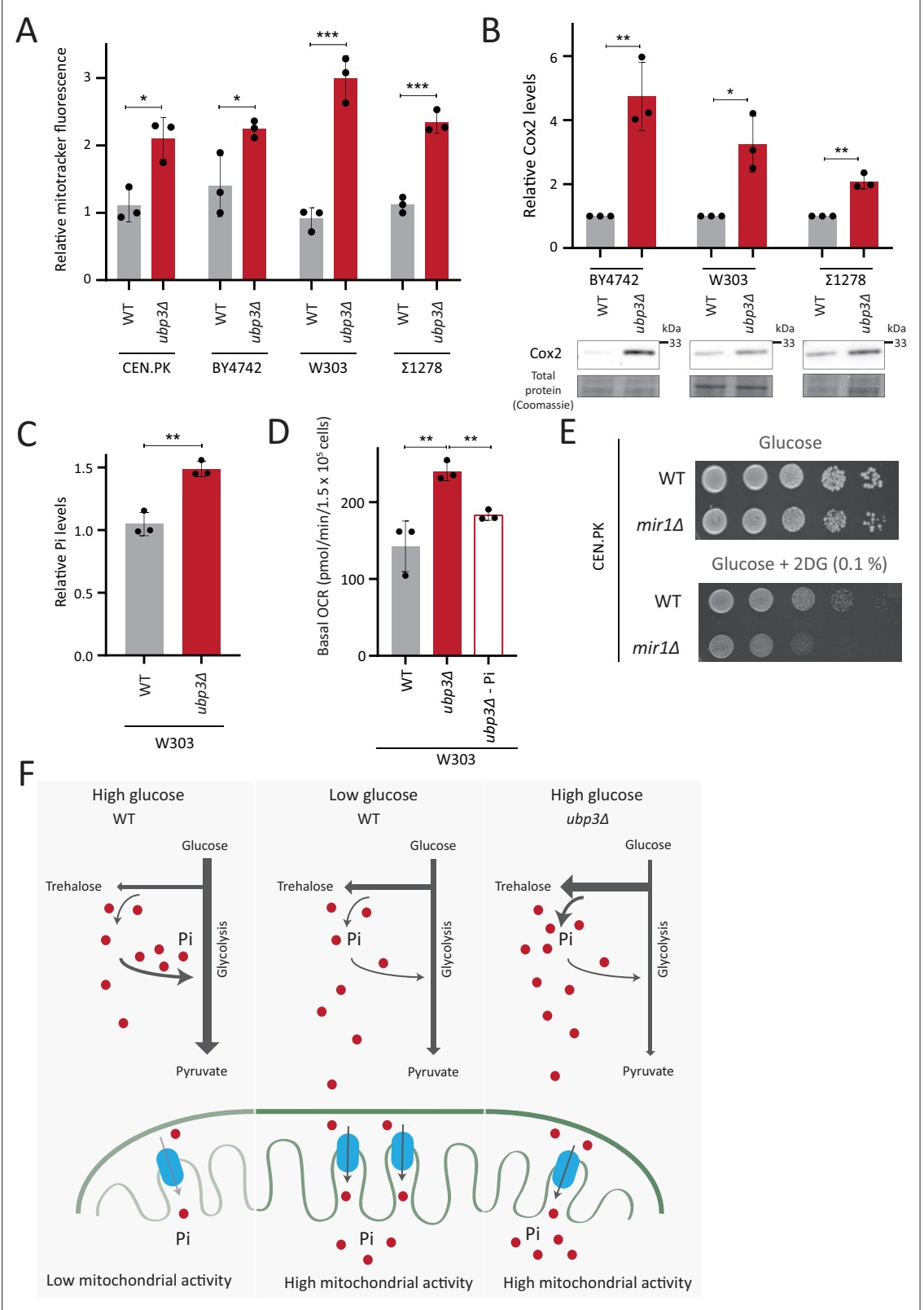

**Figure 6.** Repression of mitochondrial respiration via inorganic phosphate (Pi) budgeting is conserved in Ubp3 mutants across diverse yeast genetic backgrounds. (**A**) Effect of loss of Ubp3 on mitochondrial membrane potential in different yeast strains. Wild-type (WT) and *ubp3Δ* cells (in CEN. PK as also shown earlier in the manuscript, BY4742, W303, and Σ1278 strains of *S. cerevisiae*) were grown in high glucose and relative mitochondrial membrane potential was measured. Data represent mean ± SD from three biological replicates (n=3). (**B**) Effect of loss of Ubp3 on electron transport

*Figure 6 continued on next page*

*Figure 6 continued*

chain (ETC) complex IV subunit Cox2. WT and *ubp3Δ* (in BY4742, W303, and Σ1278 strains of *S. cerevisiae*) were grown in high glucose, and Cox2 was measured. A representative blot (out of three biological replicates, n=3) and their quantifications are shown. Data represent mean ± SD. (**C**) Intracellular Pi levels in WT and *ubp3Δ* in W303 strain background. WT and *ubp3Δ* (in W303 strain background) were grown in high glucose and the total free phosphate (Pi) levels were estimated. Data represent mean ± SD from three biological replicates (n=3). (**D**) Effect of low Pi on the basal oxygen consumption rate (OCR) in WT and *ubp3Δ* cells in W303 strain background. WT cells were grown in high glucose and *ubp3Δ* were grown in high glucose and low Pi, and basal OCR was measured. Data represent mean ± SD (n=3). (**E**) Requirement of mitochondrial Pi transport for growth after 2-deoxyglucose (2DG) treatment. Shown are serial dilution growth assays in high glucose in the presence and absence of 0.1% 2DG, using WT and *mir1Δ* cells. The results after 40 hr incubation/30°C are shown. (**F**) A model illustrating how mitochondrial Pi availability controls mitochondrial activity. In high glucose, the decreased Pi due to high Pi consumption in glycolysis, along with the glucose-mediated repression of mitochondrial Pi transporters, decreases mitochondrial Pi availability. This reduces mitochondrial activity. In low glucose, increased mitochondrial Pi transporters and lower glycolytic flux increases mitochondrial Pi, leading to enhanced mitochondrial activity. In *ubp3Δ* cells in high glucose, high trehalose synthesis and lower glycolytic flux results in an increase in Pi. This increases mitochondrial Pi availability and thereby the mitochondrial activity. Data information: *p<0.05, **p<0.01, ***p<0.001.

The online version of this article includes the following source data for figure 6:

**Source data 1.** Uncropped and labeled gels and blots for *Figure 6*.

**Source data 2.** Raw unedited gels and blots for *Figure 6*.

GAPDH. Unlike loss-of-function mutants, a reduction in amounts will only rewire metabolic flux. By 'hitting' multiple steps in glycolysis simultaneously, *ubp3Δ* have decreased Pi consumption, as well as increased Pi release. Such a cumulative phenomenon reveals more than inhibiting only GAPDH, where increased Pi comes only from reduced Pi consumption, and not from increased trehalose biosynthesis. Our serendipitous identification of a regulator which regulates multiple steps in glucose metabolism to change the metabolic environment now suggests a general basis of mitochondrial regulation that would have otherwise remained hidden. Separately, finding the substrates of Ubp3 and whether Ubp3 directly regulates glycolytic enzymes are exciting future research questions requiring concurrent innovations in accessible chemical-biological approaches to study DUBs.

Because phosphates are ubiquitous, it is challenging to identify hierarchies of Pi-dependent processes in metabolic state regulation (*Gupta and Laxman, 2021*). Phosphate transfer reactions are the foundation of metabolism, driving multiple, thermodynamically unfavorable reactions (*Kamerlin et al., 2013*; *Westheimer, 1987*). Contextually, the laws of mass action predict that the relative rates of these reactions will regulate overall Pi balance, and contrarily the Pi allocation to Pi-dependent reactions will determine reaction rates (*Gupta and Laxman, 2021*; *van Heerden et al., 2014*). Additionally, cells might control Pi allocations for different reactions via compartmentalizing Pi in organelles, to spatially restrict Pi availability (*Booth and Guidotti, 1997*; *Solesio et al., 2021*; *Vila et al., 2022*). Our data collectively suggest a paradigm where the combination of factors regulates mitochondrial Pi and thereby activity. In glycolytic yeast cells growing in high glucose, mitochondrial Pi availability becomes restricted due to higher utilization of Pi in glycolysis compared to mitochondria. Consistent with this, a rapid decrease in Pi upon glucose addition has been observed (*Hohmann et al., 1996*; *Koobs, 1972*; *Rodríguez-Enríquez et al., 2001*). Interesting, in vitro studies with isolated mitochondria from tumor cells also find that decreasing Pi levels decreases respiration (*Rodríguez-Enríquez et al., 2001*), which would be consistent with this scenario. Further, supplementing Pi correlates with decreased mitochondrial repression in tumors (*Brin and Mckee, 1956*; *Koobs, 1972*). By increasing Pi through a systems-level rewiring of glucose metabolism (such as in *ubp3Δ* cells), cells can collectively increase mitochondrial access to Pi. This Pi budgeting determines mitochondrial activity. Supplementing Pi under conditions of low glycolysis (where mitochondrial Pi transport is enhanced), as well as directly supplementing Pi to isolated mitochondria, increases respiration (*Figure 5*, *Figure 5—figure supplement 1*). Notably, this increased respiration does not happen upon directly supplementing Pi to highly glycolytic WT cells, where the Pi increases in cytosol, without increasing mitochondrial Pi (*Figure 5—figure supplement 1C*). Therefore, in order to derepress mitochondria, a combination of increased Pi along with decreased glycolysis is required. An additional systems-level phenomenon that might regulate Pi transport to the mitochondria is the decrease in cytosolic pH upon decreased glycolysis (*Dechant et al., 2010*; *Orij et al., 2011*). The cytosolic pH in highly glycolytic cells is ~7, and decreasing glycolysis results in cytosolic acidification (*Dechant et al., 2010*; *Orij et al., 2011*). Therefore, under conditions of decreased glycolysis (2DG

treatment, deletion of Ubp3, and decreased GAPDH activity), cytosolic pH becomes acidic. Since mitochondrial Pi transport itself is dependent on the proton gradient, a low cytosolic pH would favor mitochondrial Pi transport (**Hamel et al., 2004**). Therefore, under conditions of decreased glycolysis (2DG treatment, or loss of Ubp3, or decreased GAPDH activity), where cytosolic pH would be acidic, increasing cytosolic Pi might indirectly increase mitochondria Pi transport, thereby leading to increased respiration. Alternately, increasing mitochondrial Pi transporter amounts can achieve the same result, as seen by overexpressing Mir1 (**Figure 5**). A similar observation has been reported in *Arabidopsis*, reiterating an evolutionarily conserved role for mitochondrial Pi in controlling respiration (**Jia et al., 2015**). Relatedly, glycolytic inhibition can suppress cell proliferation in Warburg-positive tumors (**O'Neill et al., 2019**; **Pelicano et al., 2006**), or inflammatory responses (**Soto-Heredero et al., 2020**). However, these cells survive by switching to mitochondrial respiration (**Lu et al., 2015**; **Shiratori et al., 2019**), requiring alternate approaches to prevent their proliferation (**Cheng et al., 2012**). Inhibiting mitochondrial Pi transport in combination with glycolytic inhibition could restrict the proliferation of Warburg/Crabtree-positive cells. It is important to highlight that our experiments, whether involving Pi supplementation or Pi limitations, maintain the cellular Pi concentration within the millimolar range, and are conducted within a short timeframe (~1 hr). This differs significantly from Pi starvation studies, where cells are subjected to prolonged and complete Pi deprivation. In those contexts, cells trigger extensive metabolic adaptations in order to sustain available Pi pools including an increase in mitochondrial membrane potential which can be independent of respiration (**Ouyang et al., 2024**).

Since its discovery in the 1920s, the phenomenon of accelerated glycolysis with concurrent mitochondrial repression has been intensely researched. Yet, the biochemical constraints for glucose-mediated mitochondrial repression remains unresolved. One hypothesis suggests that the availability of glycolytic intermediates might determine the extent of mitochondrial repression. F1,6BP inhibits complex III and IV of the ETC in Crabtree-positive yeast (**Diaz-Ruiz et al., 2011**; **Díaz-Ruiz et al., 2008**; **Hammad et al., 2016**; **Rosas Lemus et al., 2018**). Similarly, the ratio between G6P and F1,6BP regulates the extent of mitochondrial repression (**Díaz-Ruiz et al., 2008**; **Rosas Lemus et al., 2018**). Although G6P/F6P accumulates in *ubp3Δ* (**Figure 2C**), this is not the case in *tdh2Δtdh3Δ* (GAPDH mutant) (**Figure 3—figure supplement 1D**), suggesting that G6P/F6P accumulation in itself is not the criterion to increase mitochondrial activity. Separately, the competition for common metabolites/co-factors between glycolysis and respiration (such as ADP, Pi, or pyruvate) could drive this phenomenon (**Diaz-Ruiz et al., 2011**; **Koobs, 1972**). Here, we observe that Mpc3-mediated mitochondrial pyruvate transport alone cannot increase respiration. An additional consideration is the possible contribution of changes in ADP in regulating mitochondrial activity, where the use of ADP in glycolysis might limit mitochondrial ADP. Therefore, when Pi changes as a consequence of glycolysis, it could be imagined that a change in ADP balance can coincidentally occur. However, prior studies show that even though cytosolic ADP decreases in the presence of glucose, this does not limit mitochondrial ADP uptake, or decrease respiration, due to the very high affinity of the mitochondrial ADP transporter (**Diaz-Ruiz et al., 2011**; **Rodríguez-Enríquez et al., 2001**). These collectively reiterate the importance of Pi access and transport to mitochondria in constraining mitochondrial respiration. Indeed, this interpretation can also contextually explain observations from other model systems where mitochondrial Pi transport seems to regulate respiration (**Scheibye-Knudsen and Quistorff, 2009**; **Seifert et al., 2015**).

We parsimoniously suggest that Pi access to the mitochondria as a key constraint for mitochondrial repression under high glucose. In a hypothetical scenario, a single-step event in evolution, reducing mitochondrial Pi transporter amounts, and/or increasing glycolytic flux (to deplete cytosolic Pi), will result in whole-scale metabolic rewiring to repress mitochondria. More elaborate regulatory events can easily be imagined as subsequent adaptations to enforce mitochondrial repression. Given the central role played by Pi, something as fundamental as access to Pi will constrain mitochondrial repression. Concurrently, the rapid incorporation of Pi into faster glycolysis can give cells a competitive advantage, while also sequestering Pi in the form of usable ATP. Over the course of evolution, this could conceivably drive other regulatory mechanisms to enforce mitochondrial repression, leading to the currently observed complex regulatory networks and signaling programs observed in the Crabtree effect, and other examples of glucose-dependent mitochondrial repression.

## Materials and methods

### Statistics and graphing

Unless otherwise indicated, statistical significance for all indicated experiments were calculated using unpaired Student's t-tests (GraphPad Prism 9.0.1). Graphs were plotted using GraphPad Prism 9.0.1.

### Yeast strains, media, and growth conditions

A prototrophic CEN.PK strain of *S. cerevisiae* (WT) (*van Dijken et al., 2000*) was used unless mentioned otherwise. Strains are listed in *Appendix 1—table 1*. Gene deletions, chromosomal C terminal-tagged strains were generated by PCR-mediated gene deletion/tagging (*Longtine et al., 1998*). Mitochondria-targeted mNeon strain (Mito-mNeon green) is described in *Dua et al., 2022*. The cox2-62 strain is described in *Bonnefoy et al., 2001*. Media compositions, growth conditions, and CRISPR-Cas9-based mutagenesis are described in Extended methods (Appendix 1).

### Mitotracker fluorescence

Mitotracker fluorescence was measured using Thermo Varioscan LUX multimode plate reader (579/599 excitation/emission). Detailed protocol is described in Extended methods (Appendix 1). Mitotracker fluorescence were normalized using $OD_{600}$ of each sample and relative fluorescence intensity calculated.

### Protein extraction and western blotting

Total protein was precipitated, extracted using TCA as described earlier (*Vengayil et al., 2019*). Blots were quantified using ImageJ software. Detailed protocol is described in Extended methods (Appendix 1).

### Basal OCR measurement

The basal OCR was measured using Agilent Seahorse XFe24 analyzer. Basal OCR readings were normalized for cell number (using $OD_{600}$ of samples) in each well. The detailed methods are described in Extended methods (Appendix 1).

### Mitochondrial volume estimation

High-resolution 3D fluorescence experiments were performed on an inverted confocal laser scanning microscope (Carl Zeiss LSM 780 or Olympus FV3000). For each imaging field of view, sequential z-stacks were acquired for each excitation channel. 488 nm laser excitation for mNeonGreen and 561 nm laser excitation for Mitotracker CMXros dye were used respectively. Images taken were deconvolved and analyzed further in ImageJ software with custom-written routines. Mitochondria segmentation and quantification was done using the Mitochondria Analyzer plugin (*Chaudhry et al., 2020*) in ImageJ. For visualization, maximum intensity projection of 3D images was used.

### RNA extraction and RT-qPCR

The RNA extraction was done using the hot phenol extraction method as described in *Vengayil et al., 2019*. The isolated RNA was DNase treated, and used for cDNA synthesis. Superscript III reverse transcriptase enzyme (Invitrogen) was used for cDNA synthesis and RT-qPCR was performed using KAPA SYBR FAST qRT PCR kit (KK4602, KAPA Biosystems). Taf10 was used as a control for normalization and the fold change in mRNA levels were calculated by $2^{-\Delta\Delta ct}$ method.

### ATP, ethanol, and Pi measurements

ATP levels were measured by ATP estimation kit (Thermo Fisher A22066). Ethanol concentration in the medium was estimated using potassium dichromate-based assay described in *Sriariyanun et al., 2019*, with modifications. Pi was estimated using a malachite green phosphate assay kit (Cayman Chemicals, 10009325). Detailed sample collection and assay protocols are described in Extended methods (Appendix 1).

### Metabolite extraction and analysis by LC-MS/MS

The steady-state levels and relative $^{13}C$ label incorporation into metabolites were estimated by quantitative LC-MS/MS methods as described in *Walvekar et al., 2018*. Detailed methodology is extensively

described in Extended methods (Appendix 1). Peak area measurements are listed in *Supplementary file 1*.

## Mitochondrial isolation

Mitochondria was isolated by immunoprecipitation as described in *Chen et al., 2017*; *Liao et al., 2018*, with modifications. The detailed protocol is described in Extended methods (Appendix 1). The eluted mitochondria were used in malachite green assay for Pi estimation, boiled with SDS-glycerol buffer for western blots or incubated with mitotracker CMXROS with mitochondrial activation buffer for mitotracker assays.

## Cytosolic fraction isolation

Cytosolic fraction was isolated from spheroplasts by centrifugation as described in detail in Appendix 1. The total protein amounts in the cytosolic fraction was estimated by BCA protein estimation assay and the Pi levels were estimated by malachite green assay.

## Acknowledgements

We thank all SL lab members, Ben Tu, Anand Bachhawat, Vijay Jayaraman for comments and suggestions. We acknowledge the NCBS/inStem/CCAMP mass spectrometry facility for LC-MS/MS support. We thank Aayushee Khanna for help with experiments and Gaurav Singh for help with microscopy. VV acknowledges funding support from DST-INSPIRE fellowship (IF170236) from the Department of Science and Technology (DST), Govt. of India, SN and SA acknowledges intramural funding from the inStem PhD program. SL acknowledges support from the DST SERB CRG grant CRG/2019/004772, a DBT-Wellcome India Alliance Senior Fellowship (IA/S/21/2/505922), and institutional support from inStem.

## Additional information

### Funding

| Funder | Grant reference number | Author |
|---|---|---|
| Wellcome Trust/DBT India Alliance | IA/S/21/2/505922 | Sunil Laxman |
| Department of Biotechnology, Ministry of Science and Technology, India | DBT SRNBIOS | Sunil Laxman |
| Department of Science and Technology, Ministry of Science and Technology, India | IF170236 | Vineeth Vengayil |
| Department of Science and Technology, Ministry of Science and Technology, India | CRG/2019/004772 | Sunil Laxman |

The funders had no role in study design, data collection and interpretation, or the decision to submit the work for publication. For the purpose of Open Access, the authors have applied a CC BY public copyright license to any Author Accepted Manuscript version arising from this submission.

### Author contributions

Vineeth Vengayil, Conceptualization, Data curation, Formal analysis, Validation, Investigation, Visualization, Methodology, Writing - original draft, Writing - review and editing; Shreyas Niphadkar, Data curation, Formal analysis, Investigation, Visualization; Swagata Adhikary, Formal analysis, Validation, Investigation; Sriram Varahan, Validation, Investigation; Sunil Laxman, Conceptualization, Resources,

Formal analysis, Supervision, Funding acquisition, Investigation, Visualization, Methodology, Writing - original draft, Project administration, Writing - review and editing

**Author ORCIDs**
Vineeth Vengayil (ID) http://orcid.org/0000-0002-1078-5476
Shreyas Niphadkar (ID) https://orcid.org/0009-0002-4028-395X
Swagata Adhikary (ID) http://orcid.org/0000-0003-2470-8356
Sunil Laxman (ID) https://orcid.org/0000-0002-0861-5080

Reviewer #2 (Public Review): https://doi.org/10.7554/eLife.90293.4.sa1
Author response https://doi.org/10.7554/eLife.90293.4.sa2

## Additional files

### Supplementary files
• Supplementary file 1. The peak areas and retention times for the metabolites (labeled and unlabeled) from the LC-MS/MS measurements are shown for wild-type (WT) and other mutants. The peak areas were analyzed by Multiquant software, version 3.0.1.

• MDAR checklist

### Data availability
All data generated or analysed during this study are included in the manuscript and supporting files; all raw data are provided as a supplementary file.

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

## Appendix 1

### Extended methods

#### Media and growth conditions

Media used in this study are high glucose (1% yeast extract, 2% peptone, and 2% glucose), low glucose (1% yeast extract, 2% peptone, and 0.1% glucose), and ethanol (1% yeast extract, 2% peptone, and 2% ethanol). The YPD-low Pi medium was prepared as previously described (*Kaneko et al., 1982*). Briefly, 1 L YP medium was prepared with 10g yeast extract and 20g peptone in 800ml water. To this, 10 ml of 1M MgS04 and 10ml aqueous ammonia were added and incubated for 30 mins at room temperature (RT), to precipitate inorganic phosphate. The precipitate was filtered and the pH of the clear solution adjusted to 5.8 using HCl, and the volume made up to 1 L. This medium was autoclaved and sterile dextrose was added to prepare the YPD-no Pi medium. The low Pi medium was prepared by adding an indicated concentration of filter-sterilized KH2PO4 solution. For experiments involving shift to low glucose, cells were grown in high glucose (2%) and at $OD_{600}$ ~0.6, cells were pelleted and shifted to low glucose (0.1%), for one hour. For experiments involving a shift to a no-Pi or low glucose medium, cells were subcultured in high glucose till $OD_{600}$~0.6, pelleted by centrifugation (1000 x g, 2 min at RT), washed, and shifted to no-Pi or low glucose medium for 1 hour. For experiments involving measurement of basal OCR at different Pi concentrations (*Figure 5—figure supplement 1*), cells were grown in 5 mM Pi medium (prepared by removing Pi from YPD medium and adding $KH_2PO_4$ at the required concentration) and at $OD_{600}$ ~0.6, the cells were supplemented with additional Pi at required final concentrations. MG132 was used at a final concentration of 100 μM and the experiments involving MG132 addition were performed in pdr5Δ background to prevent MG132 efflux by Pdr5 transporter. Experiments involving heat stress were performed by incubating cells at 42°C for one hour.

#### CRISPR-Cas9 based mutagenesis

To generate the Ubp3$^{C469A}$ strain, cells were transformed a constitutive Cas9 expressing plasmid (Addgene plasmid 43802, *DiCarlo et al., 2013*). Guide RNA (gRNA) sequences (forward and reverse) were annealed and cloned into a pMEL13 plasmid. Strains were then transformed with gRNA plasmid and homology repair (HR) fragments with the base pair mutation. The HR fragment was generated by site-directed mutagenesis PCR. Positive clones were selected using drug, and sequenced to confirm the mutation. Oligonucleotides used are listed in *Appendix 1—table 2*.

#### Sample preparation for mitotracker fluorescence assay

Cells were grown to OD600~0.6, 1 OD600 cells were collected by centrifugation (1000 x g, 1 min at RT), and resuspended in 1 ml fresh media. Mitotracker CMXRos (M7512 ThermoFisher) was added to a final concentration of 200 nM and incubated at 30°C in a shaking incubator. Cells were washed and resuspended in fresh media and fixed with 2% formaldehyde for 20 minutes. Fixed cells were washed in 1xPBS, pH 7.4, and resuspended in 1 ml 1xPBS. 300 μl of this was aliquoted onto 96 well plates in replicates.

#### Mitotracker based screen to identify DUBs that regulate mitochondrial repression

The deubiquitinase deletions were made by PCR mediated gene deletion, DUB deletion mutants generated were grown in high glucose and the mitochondrial membrane potential measured by the mitotracker assay described earlier. For performing the screen, the 19 DUB knockouts were divided into two groups- each containing 10 and 9 DUB KOs respectively and the screen was performed separately for each group in replicates. WT cells grown in high glucose indicated the basal mitochondrial potential and WT cells grown in ethanol was used as a positive control for each group. The mitotracker fluorescence intensity were normalised to OD600 of each mutant and the fluorescence intensity relative to the WT cells were calculated. The relative fluorescence intensity for the replicate samples are shown in *Figure 1—figure supplement 1A*, mean fluorescence intensities of the replicates (relative to WT) were calculated and plotted as heat maps (*Figure 1C*).

#### Serial dilution growth assay

For serial dilution-based growth assays, cells were grown in standard high glucose to OD600~0.8, collected by centrifugation (1000 x g, 1 min at RT), washed with water, serial dilutions made

(OD600=1, 0.1, 0.01, 0.001, 0.0001), 5 µl of each dilution were spotted onto agar plates, incubated at 30°C and growth monitored.

## Protein extraction and western blotting
The cells were grown in the indicated medium to OD600~0.8 and pelleted by centrifugation(1000 x g, 2 min at RT). Total protein was precipitated and extracted using trichloroacetic acid (TCA), and resuspended in SDS/glycerol buffer. The supernatant was collected after centrifugation, and total proteins were estimated by BCA assay (BCA assay kit, G-Biosciences). Protein samples were normalized to ensure the same protein amounts in all the samples, in SDS/glycerol buffer. Samples were resolved on 4–12% bis-tris gels (Invitrogen, NP0336BOX), using MOPS running buffer (50 mM tris, 50 mM MOPS, 0.1% SDS, and 1 mM EDTA), the gels cut so that the relevant portion of the gel is transferred to a nitrocellulose membrane (GE Healthcare, 10600003), while a lower or higher region was stained with coomassie blue for protein normalization, and blots were developed using the following antibodies: anti-HA mouse (Sigma-Aldrich 11583816001), anti-FLAG mouse (Sigma-Aldrich F1804), anti-Cox2 mouse (Invitrogen MTCO2 459150), anti-ubiquitin (P4D1 mouse mAb, CST), anti-Idh1 goat (Sigma-Aldrich SAB2501682). Horseradish peroxidase-conjugated secondary antibody was from Sigma-Aldrich (mouse and rabbit) and Thomas scientific (goat). Chemiluminescence was detected by using Western Bright ECL HRP substrate, Advansta, K12045.

## Seahorse assay
1 ml of the XF Calibrant solution was aliquoted into each well of the utility plate and the sensor cartridge plate was hydrated overnight as per the manufacturer's instructions. The sensor cartridge was loaded with complex IV inhibitor sodium azide and equilibrated in the Seahorse XFe24 analyzer one hour prior to the start of the assay. The culture plate was coated with 50 µl poly-L-lysine and incubated for 1 hour at RT. Excess poly-L-lysine was removed and the plate dried at 30°C for 30 minutes. The cells were grown in standard high glucose to an OD600~0.6. The samples were aliquoted to poly-L-lysine coated wells at a final cell number in each well of ~3x10^5. The plate was centrifuged for 2 min at 100xg (acceleration 2, brake 2) and incubated at 30°C for 30 minutes. The plate was loaded to the Seahorse XFe24 analyzer and basal OCR measured over time. A minimum of three measurements were taken with intermittent 2 min mixing and waiting steps. This was followed by sodium azide injection to the sample wells, and three measurements were taken with intermittent 2 min mixing and waiting steps.

## Sample preparation for ATP estimation assay
Cells were grown in high glucose and at $OD_{600}$ ~0.8, 10 $OD_{600}$ were pelleted at 4°C. The pellet was treated with 300 µl ice-cold TCA (5%), resuspended and incubated in ice for 15 mins. The suspension was diluted such that the final TCA concentration is 0.1% using 20 mM Tris-HCl, Ph 7.0. For calculating the contribution of mitochondria to total ATP, cells were grown in high glucose and at $OD_{600}$ ~0.6, sodium azide was added at a final concentration of 1 mM and incubated for 45 minutes at 30°C in a shaking incubator. The reaction mixture for ATP measurements were prepared as per manufacturer's instructions and luminescence was measured using Sirius luminometer (Tiertek Berthold). The relative ATP concentrations were calculated, graphs were plotted and using GraphPad prism 9.0.1.

## Sample preparation for ethanol estimation
Cells were grown in appropriate media conditions to an OD600~1, 10 ml of cells collected, and centrifuged at 3000 x g at 4°C. 5 ml of the supernatant was collected in a fresh tube and 1ml of Tri-n-butyl phosphate (TBP) was added. The mixture was vortexed for 5 minutes and centrifuged at 3420xg for 5 minutes to separate the phases. 500 µl of the top, clear phase was transferred to a fresh tube and 500 µl of potassium dichromate reagent was added (10% w/v of $K_2Cr_2O_7$ in 5 M of $H_2SO_4$), followed by vortexing for 1 minute. The mixture was incubated at RT for 10 minutes and 200 µl of the bottom layer was transferred to a 96 well plate. The absorbance in each well was measured at 595 nm. A standard curve was plotted using medium containing different concentrations of added ethanol, and was used to calculate the ethanol concentration in samples. For calculating the rate of ethanol production, WT and ubp3Δ cells were grown in high glucose and at OD600~0.6, equal number of cells were shifted to fresh high glucose medium. The supernatant from 10ml cells were collected, and ethanol concentration in the medium was estimated at different time points after the

shift (normalised to $OD_{600}$ at each time point, to consider the growth difference between WT and ubp3Δ cells).

## Sample preparation for Pi estimation by malachite green assay

The cells were grown to OD600~0.8, pelleted by centrifugation (1000 x g, 2 min at RT), pelleted cells were washed and resuspended in 1 ml ice-cold HPLC grade water (to avoid Pi contamination) and lysed by bead beating (3 x 20s, 1 min intermittent cooling). Lysates were centrifuged (14000 rpm, 1 min at 4°C) and 800 µl of the clear supernatant collected. Protein concentrations of the supernatant were estimated using BCA assay (BCA assay kit, GBiosciences). Lysates corresponding to 1µg total protein was used for the Pi estimation assay.

## Metabolite extraction and analysis by LC-MS/MS

### Steady state measurements of glycolytic intermediates

For measuring steady-state levels of glycolytic intermediates, trehalose, and PPP intermediates, cells were grown in standard high glucose medium to an $OD_{600}$~0.8, 5 $OD_{600}$ cells were collected, quenched in 60% methanol at -40°C, and metabolites were extracted as described in *Walvekar et al., 2018*. The extracted metabolites were separated by Synergi 4µm Fusion-RP 80 Å LC column (150 x 4.6 mm, Phenomenex) on Waters Acquity UPLC system, and measured as described earlier (*Walvekar et al., 2018*), in negative polarity mode. Solvents used: 5 mM ammonium acetate in water (solvent A) and 100% acetonitrile (solvent B). ABSciex QTRAP 6500 mass spectrometer was used and data acquisition was done usingAnalyst 1.6.2 (Sciex). Detailed flow parameters are described elsewhere (*Walvekar et al., 2018*). MultiQuant version 3.0.1 was used for data analysis. The parent and product ion masses used for the analysis are listed in *Appendix 1—table 3*. The peak areas for individual metabolites were calculated and values were plotted relative to WT. The retention time for different metabolites and peak area obtained after analysis are listed in *Supplementary file 1*. The statistical significance was calculated using unpaired Student's t-test (GraphPad prism9.0.1).

### Measurement of $^{13}$C carbon flux into glycolysis and trehalose

The carbon flux into glycolysis and trehalose was estimated by measuring the relative incorporation of $^{13}$C label into glycolytic intermediates and trehalose after a pulse of $^{13}$C$_6$ glucose. Cells were grown in high glucose medium with 1% glucose and at $OD_{600}$~0.6, the cells were shifted to fresh medium with 1% glucose. The $OD_{600}$ was measured 30 minutes after the shift and cells corresponding to ~5 $OD_{600}$ were collected, and $^{13}$C$_6$ glucose (Cambridge Isotope Laboratories, CLM-1396) was pulsed at a final concentration of 1%, making the total glucose concentration in the medium 2% (1% labelled + 1% unlabelled). Note: Since label saturation into glycolytic intermediates happens instantaneously (within seconds) after $^{13}$C glucose pulse, the time points for sample collection should be determined prior to the experiment by plotting total label percentage over time for individual metabolites to calculate the time point at which label saturation takes place. For measuring $^{13}$C label incorporation, the cells were collected at 3s (for F1,6BP, G3P, 3PG and PEP), 10s (for G6P/F6P) and 4 mins (for trehalose) after $^{13}$C$_6$ glucose pulse, based on the percentage label incorporation plotted in figure S2C. The cells were quenched in 60% methanol at -40°C, and metabolites were extracted as described in *Walvekar et al., 2018*. The extracted metabolites were detected as described earlier. The relative $^{13}$C label incorporation in ubp3Δ cells were measured by calculating the area under the peak for individual $^{13}$C labelled metabolites (using MultiQuant version 3.0.1) and dividing it by the peak area of that specific metabolite in the WT sample. The change in $^{13}$C labelled metabolites relative to WT samples were plotted. For calculating the total $^{13}$C label percentage of a metabolite, the peak area of $^{13}$C labelled metabolites were divided by the sum of peak areas of that metabolite (labelled plus unlabelled) and expressed as percentage. The total label percentage was plotted over time for individual metabolites to calculate the time point at which label saturation takes place (Figure S2C). The retention time for different labelled and unlabelled metabolites and peak area obtained after analysis are listed in *Supplementary file 1*. The statistical significance was calculated using unpaired Student's t-test (GraphPad prism 9.0.1).

### Sample preparation for cytosolic fraction separation and mitochondria isolation by immunoprecipitation

Cells with Tom20 (for mitochondria isolation) and Eno1 (for cytosolic fraction isolation) endogenously tagged at the C terminus with a 3X FLAG epitope tag were grown in 300 ml high glucose medium and at $OD_{600}$~ 0.8, the cells were pelleted at 1500 x g for 5 minutes at 4°C. The cells were washed

in water, pelleted and weighed. The cells were then incubated in Tris-DTT buffer (0.1 M Tris-SO$_4$, 10 mM DTT, pH 9.4) (5 ml/g pellet weight) for 15 mins at 30°C in a shaking incubator. The cells were washed in SEH buffer (0.6 M sorbitol, 20 mM HEPES-KOH, 2mM MgCl$_2$, pH 7.4) (5 ml/g pellet weight) and incubated with Zymolyase 20T (0832092, MP biomedicals) dissolved in SEH buffer for 60 minutes, at 30°C in a shaking incubator. The spheroplasts produced by Zymolyase treatment were collected by centrifugation at 4500 x g for 5 mins at 4°C, washed with ice-cold SEH buffer (5 ml/g pellet weight) and resuspended in ice-cold SEH buffer with protease inhibitors (1 mM PMSF, 0.5 mg/ml pepstatin A and 0.5 µg/ml leupeptin) (5 ml/g pellet weight). The resuspended spheroplasts were homogenised using a Dounce homogeniser, centrifuged at 1500 x g for 5 mins and the supernatant was collected. The supernatant was centrifuged at 12000 x g for 10 mins at 4°C. The supernatant cytosolic fraction was collected and the pellets were resuspended in 700 µl ice-cold SEH buffer to obtain the mitochondria-enriched fraction. To purify mitochondria by immunoprecipitation, the mitochondria-enriched fraction was incubated with 1.5 mg Dynabeads protein G (Invitrogen) conjugated with 10 µg of anti-FLAG antibody (mouse, Sigma-Aldrich F1804), for 60 minutes at 4°C. The beads were separated using a magnet, washed 3 times in ice-cold SEH buffer and the mitochondria were eluted in 200 ul SEH buffer with FLAG peptide (1.5 mg/ml).

## Mitotracker assay using isolated mitochondria

For mitotracker assays with isolated mitochondria (*Figure 5D*), the mitochondria enriched fraction resuspended in SEH buffer was centrifuged twice- 700 x g for 5mins and 1500 x g for 5 mins at 4°C. The supernatant was collected and centrifuged at 12000 x g for 10 mins at 4°C and the pellet was resuspended in ice-cold SEH buffer. The protein concentration of this crude mitochondria was estimated by BCA protein estimation assay and mitochondria corresponding to 5 µg protein was incubated in 200 µl SEH buffer containing 1 mM pyruvate, 1 mM malate, 0.5 mM ADP and 0 – 50 mM KH$_2$PO$_4$ for 30 minutes at 30°C. Mitotracker CMXROS was added at a final concentration of 200 nM, followed by incubation at 30°C for 30 minutes. The fluorescence intensities were measured using Thermo Varioscan LUX multimode plate reader at 579/599 excitation/emission, and fluorescence intensities relative to samples containing 0 mM KH2PO4 was calculated. Statistical significance was calculated using unpaired Student's t-test (GraphPad prism 9.0.1).

**Appendix 1—table 1.** List of strains used in this study.

| SL no. | Strain name | Genotype | Description | Source |
|---|---|---|---|---|
| 1 | CEN.PK a (WT) | Mat a | haploid strain of CEN.PK MAT 'a' mating type | *van Dijken et al., 2000* |
| 2 | DUBs KO strains | CEN.PK a::hphMX6 | Deletion of the individual deubiquitinases (see *Figure 1—figure supplement 1A*) | This study |
| 3 | Ubp3C469A | CEN.PK a UBP3C469A | Catalytically inactive Ubp3 mutant | This study |
| 4 | Δatp1 | CEN.PK a Δatp1::natMX6 | Deletion of ATP1 gene | This study |
| 5 | Δatp1 Δubp3 | CEN.PK a Δubp3::hphMX6 Δatp1:: natMX6 | Deletion of ATP1 gene in a UBP3 deletion strain | This study |
| 6 | Δatp10 | CEN.PK a Δatp10::natMX6 | Deletion of ATP10 gene | This study |
| 7 | Δatp10 Δubp3 | CEN.PK a Δubp3::hphMX6 Δatp10:: natMX6 | Deletion of ATP10 gene in a UBP3 deletion strain | This study |
| 8 | Pfk1-FLAG | CEN.PK a PFK1 3x FLAG::natMX6 | C terminal tagged Pfk1 | This study |
| 9 | Pfk1-FLAG Δubp3 | CEN.PK a PFK1-3x FLAG::natMX6 Δubp3::hphMX6 | C terminal tagged Pfk1 in a UBP3 deletion strain | This study |
| 10 | Tdh2-FLAG | CEN.PK a TDH2-3x FLAG::natMX6 | C terminal tagged Tdh2 | This study |
| 11 | Tdh2-FLAG Δubp3 | CEN.PK a TDH2-3x FLAG::natMX6 Δubp3::hphMX6 | C terminal tagged Tdh2 in a UBP3 deletion strain | This study |

*Appendix 1—table 1 Continued on next page*

*Appendix 1—table 1 Continued*

| SL no. | Strain name | Genotype | Description | Source |
|---|---|---|---|---|
| 12 | Tdh3-FLAG | CEN.PK a TDH3-3x FLAG::natMX6 | C terminal tagged Tdh3 | This study |
| 13 | Tdh3-FLAG Δubp3 | CEN.PK a TDH3-3x FLAG::natMX6 Δubp3::hphMX6 | C terminal tagged Tdh3 in a UBP3 deletion strain | This study |
| 14 | Eno1-FLAG | CEN.PK a ENO1-3x FLAG::natMX6 | C terminal tagged Eno1 | This study |
| 15 | Eno1-FLAG Δubp3 | CEN.PK a ENO1-3x FLAG::natMX6 Δubp3::hphMX6 | C terminal tagged Eno1 in a UBP3 deletion strain | This study |
| 16 | Eno2-FLAG | CEN.PK a ENO2-3x FLAG::natMX6 | C terminal tagged Eno2 | This study |
| 17 | Eno2-FLAG Δubp3 | CEN.PK a ENO2-3x FLAG::natMX6 Δubp3::hphMX6 | C terminal tagged Eno2 in a UBP3 deletion strain | This study |
| 18 | Δtdh2 Δtdh3 | CEN.PK a Δtdh2::natMX6 Δtdh3:: kanMX6 | Deletion of TDH2 gene in a TDH3 deletion strain | This study |
| 19 | Δtps2 | CEN.PK a Δtps2::hphMX6 | Deletion of TPS2 gene | This study |
| 20 | Δtps2 Δubp3 | CEN.PK a Δtps2::hphMX6 Δubp3::natMX6 | Deletion of UBP3 gene in a TPS2 deletion strain | This study |
| 21 | Mir1-HA | CEN.PK a MIR1-6xHA::natMX6 | C terminal tagged Mir1 | This study |
| 22 | Mir1-HA Δubp3 | CEN.PK a MIR1-6xHA::natMX6 Δubp3::hphMX6 | C terminal tagged Mir1 in a UBP3 deletion strain | This study |
| 23 | Pic2-HA | CEN.PK a PIC2-6xHA::natMX6 | C terminal tagged Pic2 | This study |
| 24 | Pic2-HA Δubp3 | CEN.PK a PIC2-6xHA:: natMX6 Δubp3::hphMX6 | C terminal tagged Pic2 in a UBP3 deletion strain | This study |
| 25 | Δmir1 | CEN.PK a Δmir1::natMX6 | Deletion of Mir1 | This study |
| 26 | Δmir1 Δubp3 | CEN.PK a Δmir1::natMX6 Δubp3::hphMX6 | Deletion of Mir1 in a UBP3 deletion strain | This study |
| 27 | WT+ Empty vector | CEN.PK a pG6PD:: kanMX6 | haploid strain of CEN.PK MAT 'a' with an empty vector with kanamycin resistance | This study |
| 28 | Mir1-HA OE | CEN.PK a pG6PD-Mir1-6xHA:: kanMX6 | Mir1-HA overexpression under the constitutive G6PD promoter | This study |
| 29 | Mpc3-FLAG | CEN.PK a MPC3-3xFLAG::natMX6 | C terminal tagged Mpc3 | This study |
| 30 | Mpc3-FLAG Δubp3 | CEN.PK a MPC3-3xFLAG:: natMX6 Δubp3::hphMX6 | C terminal tagged Mpc3 in a UBP3 deletion strain | This study |
| 31 | Mpc3-FLAG Mir1-HA OE | CEN.PK a MPC3-3xFLAG:: natMX6 pG6PD-Mir1-6xHA:: kanMX6 | C terminal tagged Mpc3 in Mir1-HA overexpression | This study |
| 32 | Δmpc3 | CEN.PK a Δmpc3::natMX6 | Deletion of Mpc3 | This study |
| 33 | Tom20-FLAG | CEN.PK a Tom20-3xFLAG::natMX6 | C terminal tagged Tom20 | This study |
| 34 | Tom20-FLAG Vph1-HA | CEN.PK a Tom20-3xFLAG:: natMX6 Vph1-6xHA:: hphMX6 | C terminal tagged Vph1 in c terminal tagged Tom20 | This study |
| 35 | Mito-Mneon green | CEN.PK a HO::PCYC1-SU9m Neongreen-TCYC1-KanMX6 | Mneon gene with a mitochondria targeted sequence at the N terminus | *Dua et al., 2022* |

*Appendix 1—table 1 Continued on next page*

*Appendix 1—table 1 Continued*

| SL no. | Strain name | Genotype | Description | Source |
|---|---|---|---|---|
| 36 | cox2-62 | leu2 Δarg8 ΔURA3 ura3-52 kar1-1 ade2-101 | cox2-62 $\rho$ +, Cox2 with deletion of -295 to +363 relative to AUG | *Bonnefoy et al., 2001* |
| 37 | W303 | MAT a leu2-3,-112;his3-11,-15;trp11;ura3-1; ade2-1;can1-100 | WT W303 strain | *Ralser et al., 2012* |
| 38 | W303 Δubp3 | MAT a leu2-3,-112;his3-11,-15;trp11;ura3-1; ade2-1;can1-100Δubp3::hphMX6 | Deletion of Ubp3 in W303 | This study |
| 39 | BY4742 | MAT α his3Δ1:leu2Δ0:lys2Δ0: MET15:ura3Δ0 | WT BY4742 strain | *Winston et al., 1995* |
| 40 | BY4742 Δubp3 | MAT α his3Δ1:leu2Δ0:lys2Δ0: MET15:ura3Δ0 Δubp3::hphMX6 | Deletion of Ubp3 in BY4742 | This study |
| 41 | Σ1278 | MAT a | WT Σ1278 strain | Isolate via Fink lab |
| 42 | Σ1278 Δubp3 | MAT a ura3-52 Δubp3::hphMX6 | Deletion of Ubp3 in BY4742 | This study |
| 43 | S288C | MAT a | WT S288C strain | This study |
| 44 | Rho0 | CEN.PK Mat a | Lacks mitochondrial DNA, generated using EtBr treatment | This study |
| 45 | Rho0 Δubp3 | CEN.PK Mat a | Deletion of Ubp3 in Rho0 strain | This study |

**Appendix 1—table 2.** Oligonucleotides used for CRISPR-Cas9 based mutagenesis.

| | |
|---|---|
| Ubp3C469A gRNA forward | AGAACTCATAAAACAAATGTgtttt |
| Ubp3C469A gRNAreverse | ACATTTGTTTTATGAGTTCTgatca |
| HR fragment Ubp3C469A forward | CAAAATACCAGTCCATTCCATTATTCCAAGAGGCATAATTAACAGAGCCAAC ATTGCTTTTATGAGTTCT |
| HR fragment Ubp3C469A reverse | ACGTTAATTACATCAATAAATGGCTTACAGTAGAGTAACACTTGTAACACAGAACT CATAAAAGCAATGT |

**Appendix 1—table 3.** List of parent ion mass and product ion mass (Q1/Q3) used for detection of metabolites.

| Metabolite | Parent Ion (Q1) mass | Product Ion (Q3) mass | Collision energy (V) | Retention time |
|---|---|---|---|---|
| Glucose 6-phosphate (G6P) / Fructose 6-phosphate (F6P) | 259 | 97 | -20 | 2.68 |
| 13C_G6P/F6P_6 | 265 | 97 | -20 | 2.68 |
| Fructose 1,6-bisphosphate (F16BP) | 339 | 97 | -20 | 2.42 |
| 13C_F16BP_6 | 345 | 97 | -20 | 2.42 |
| Trehalose | 341.3 | 179.3 | -17 | 3.85 |
| 13C_Trehalose_6 | 347.3 | 185.3 | -17 | 3.85 |
| 13C_Trehalose_12 | 353.3 | 185.3 | -17 | 3.85 |
| Ribose 5-phosphate (R5P) | 229 | 97 | -20 | 2.69 |
| Sedoheptulose-7-phosphate (S7P) | 289 | 97 | -20 | 2.68 |
| Glyceraldehyde 3-phosphate (G3P) | 169 | 97 | -20 | 2.65 |
| 13C_G3P_3 | 172 | 97 | -20 | 2.65 |
| Phosphoenol pyruvate (PEP) | 167 | 79 | -12 | 2.45 |

*Appendix 1—table 3 Continued on next page*

*Appendix 1—table 3 Continued*

| Metabolite | Parent Ion (Q1) mass | Product Ion (Q3) mass | Collision energy (V) | Retention time |
|---|---|---|---|---|
| 13C_PEP_3 | 170 | 79 | -12 | 2.45 |
| 3-phosphoglycerate (3PG) | 185 | 97 | -20 | 2.53 |
| 13C_3PG_3 | 188 | 97 | -20 | 2.53 |
| Pyruvate | 299.1 | 91.1 | 28 | 8.89 |
| Citrate | 508 | 385 | 7 | 8.04 |
| 13C_ Citrate _2 | 510 | 387 | 7 | 8.03 |
| 13C_ Citrate _3 | 511 | 388 | 7 | 8.03 |
| 13C_ Citrate _4 | 512 | 389 | 7 | 7.69 |
| 13C_ Citrate _5 | 513 | 390 | 7 | 7.91 |
| 13C_ Citrate _6 | 514 | 391 | 7 | 8.42 |
| 2-Ketoglutarate (2-KG) | 462 | 339 | 11 | 8.68 |
| 13 C_2-KG_1 | 463 | 340 | 11 | 8.67 |
| 13 C_2-KG_2 | 464 | 341 | 11 | 8.67 |
| 13 C_2-KG_3 | 465 | 342 | 11 | 8.63 |
| 13 C_2-KG_4 | 466 | 343 | 11 | 8.6 |
| 13 C_2-KG_5 | 467 | 344 | 11 | 8.69 |
| Succinate | 329 | 206 | 15 | 8.21 |
| 13C_Succinate_1 | 330 | 207 | 15 | 7.32 |
| 13C_Succinate_2 | 331 | 208 | 15 | 7.32 |
| 13C_Succinate_3 | 332 | 209 | 15 | 6.97 |
| 13C_Succinate_4 | 333 | 210 | 15 | 6.96 |
| Fumarate | 327 | 91.2 | 34 | 7.55 |
| 13C_Fumarate_1 | 328 | 91.2 | 34 | 7.74 |
| 13C_Fumarate_2 | 329 | 91.2 | 34 | 7.32 |
| 13C_Fumarate_3 | 330 | 91.2 | 34 | 7.32 |
| 13C_Fumarate_4 | 331 | 91.2 | 34 | 7.32 |
| Malate | 345 | 91.2 | 33 | 7.09 |
| 13C_Malate_1 | 346 | 91.2 | 33 | 7.08 |
| 13C_Malate_2 | 347 | 91.2 | 33 | 7.08 |
| 13C_Malate_3 | 348 | 91.2 | 33 | 7.2 |
| 13C_Malate_4 | 349 | 91.2 | 33 | 6.61 |
| Oxaloacetate | 448 | 325 | 10 | 9.92 |
| 13C_Oxaloacetate_1 | 449 | 326 | 10 | 8.67 |
| 13C_Oxaloacetate_2 | 450 | 327 | 10 | 8.67 |
| 13C_Oxaloacetate_3 | 451 | 328 | 10 | 8.63 |
| 13C_Oxaloacetate_4 | 452 | 329 | 10 | 8.6 |

