## [Editor Report · eLife assessment]

This study provides **valuable** insights into the regulation of metabolic flux between glycolysis and respiration in yeast, particularly focusing on the role of inorganic phosphate. The authors propose a novel mechanism involving Ubp3/Ubp10 that potentially mitigates the Crabtree effect, offering substantial, **solid** evidence through a variety of well-designed assays. This study could reshape our understanding of metabolic regulation with broad biological contexts.

---

## [Referee Report · Reviewer #2 (Public Review)]

Summary:

Cells cultured in high glucose tend to repress mitochondrial biogenesis and activity, a prevailing phenotype type called Crabtree effect that observed in different cell types and cancer. Many signaling pathways have been put forward to explain this effect. Vengayil et al proposed a new mechanism involved in Ubp3/Ubp10 and phosphate that controls the glucose repression of mitochondria. The central hypothesis is that ∆ubp3 shift the glycolysis to trehalose synthesis, therefore lead to the increase of Pi availability in the cytosol, then mitochondrial received more Pi and therefore the glucose repression is reduced.

Strengths:

The strength is that the authors used an array of different assays to test their hypothesis. Most assays were well-designed and controlled.

Weaknesses:

The author addressed my major concerns.

---

## [Author Response]

The following is the authors’ response to the previous reviews.

**Public Reviews:**

**Reviewer #1 (Public Review):**
The study by Vengayil et al. presented a role for Ubp3 for mediating inorganic phosphate (Pi) compartmentalization in cytosol and mitochondria, which regulates metabolic flux between cytosolic glycolysis and mitochondrial processes. Although the exact function of increased Pi in mitochondria is not investigated, findings have valuable implications for understanding the metabolic interplay between glycolysis and respiration under glucose-rich conditions. They showed that UBP3 KO cells regulated decreased glycolytic flux by reducing the key Pi-dependent-glycolytic enzyme abundances, consequently increasing Pi compartmentalization to mitochondria. Increased mitochondria Pi increases oxygen consumption and mitochondrial membrane potential, indicative of increased oxidative phosphorylation. In conclusion, the authors reported that the Pi utilization by cytosolic glycolytic enzymes is a key process for mitochondrial repression under glucose conditions.Comments on revised version:This reviewer appreciates the author's responses addressing some of the concerns.(1) However, the concern of reproducibility and experimental methods applied to the study is still valid, particularly considering that many conclusions were drawn from western blot analysis. The authors used separate gel loading controls for western blot analysis, which is not a valid method. Considering loading and other errors/discrepancies during the transfer phase of the assay, the direct control should be analyzing the membrane after transfer or using an internal control antibody on the same membrane. None of the western blots are indicated with marker sizes, and it isn't very clear how many repeats there are and whether those repeats are biological or technical repeats.

We thank the reviewer for raising this concern. This point requires detailed clarification regarding two key points: the first one regarding the use of Coomassie stained gels over internal ‘housekeeping gene’ antibodies, and the second one regarding the challenges in performing controls for western blots In case of high abundance proteins such as glycolytic enzymes.

(1) In our western blots, we have used Coomassie stained gel as a loading control for all our western blots. This is performed by cutting one half of the gel and using it for transfer followed by blotting and using the other half for Coomassie staining. I.e. This is not two separate gels that are loaded, but the same gel. Practically, this is no different from cutting a membrane to blot with different antibodies. This method is of course valid method for normalizing western blot data, and is used by multiple studies, for the reasons mentioned below. The historical use of a ‘house-keeping’ gene as a loading control for western blotting *assumes* that the protein levels of these does not change under different conditions. However, this approach has multiple, severe limitations (since a ‘housekeeping gene’ is entirely contextual, and indeed), and therefore it is correct to use *total protein* as a loading control. This is indeed recommended for use by multiple studies (Collins et al., 2015). Coomassie staining for total protein is far more reliable than using house-keeping genes as a loading control in western blots (Welinder and Ekblad, 2011). A notable example would be GAPDH itself, which is widely used as a loading control in many studies. As is clear from our data in this manuscript, GAPDH levels itself decrease in *ubp3Δ* cells. Had we used GAPDH as a loading control, we wouldn’t have identified the decrease in glycolytic enzymes in *ubp3Δ* cells, and this story would have met with a tragic fate very early on in its inception. We have in fact be very careful with these quantitations, and even before loading samples on gels, they are first normalized using a standard protein estimation assay (Bradford), followed by normalized loading, followed by cutting the gel into two parts - one for coomassie staining and protein normalization, and the other for the western blot for the respective proteins. However, in point (2) below, we clarify on why sometimes we have to load a separate gel with normalized protein, which should resolve this point.

(2) Glycolytic enzymes are highly abundant proteins and to achieve a signal in the linear range of western blot, the protein extracts have to be diluted (up to 25 or 50 times). As discussed under point 1, an internal control ‘housekeeping gene’ antibody is not a reliable method to use as loading control. Even if we want to use an antibody for an internal protein as a control, there are not many proteins that are as abundant as metabolic enzymes and because of this simple reason, the sample dilution results in these proteins not getting detected in the western blot since the signal will be below the limit of detection. This leaves using a separate gel loading control as the only easy to perform, reliable option.

We would like to further highlight the fact that the changes in metabolic enzymes and ETC proteins that we observe in the ubp3 mutant by western blot, were also independently observed by large scale untargeted quantitative proteomics study by (Isasa et al., 2015), which we cite extensively in this manuscript. Since an entirelyindependent study, using a completely different (untargeted) method has also shown very similar changes in proteins that we observe (mitochondrial, and glycolytic enzymes), there should be no room for doubt regarding the altered glycolytic enzyme and ETC protein levels that we discover in this study.

None of the western blots are indicated with marker sizes

We have clearly indicated the marker sizes in all our western blots. Separately, raw images of the blots and Coomassie stained gels have been provided with the manuscript raw data, and is therefore easily available for any interested reader.

It isn't very clear how many repeats there are and whether those repeats are biological or technical repeats.We have already clearly indicated the details of each blot in the figure legends. For example *“A representative blot (out of three biological replicates, n=3) and their quantifications are shown. Data represent mean ± SD.”* We kindly request the reviewer to thoroughly go through the figure legends for details regarding the western blots, or any other data. We hope this addresses all the reviewer concerns regarding the credibility of our western blot results and the method of using Coomassie stained gels as loading controls in this study.(2) Concern regarding citing the Ouyang et al. paper is still valid. This paper is an essential implication in phosphate metabolism and is directly related to some of the findings associated with mitochondrial function, along with conflicting results, which should be discussed in the discussion section. As a reviewer, I do not request citing any paper from the authors in general; however, considering some of the conflicting results here, citing and discussing paper from Ouyang et al. will improve the interoperation/value of their findings.

As mentioned in detail in our previous response letter, we do not believe that the study from Ouyang et al., present ‘conflicting results’ of any kind. Nevertheless, in response to the reviewer's suggestion, we have revised the discussion section of our manuscript and added a few points that incorporate the insights from Ouyang et al. These are in the discussion section (“It is important to highlight that our experiments, whether involving Pi supplementation or Pi limitations, maintain the cellular Pi concentration within the millimolar range and are conducted within a short timeframe (~ 1 hour). This differs significantly from Pi starvation studies, where cells are subjected to prolonged and complete Pi deprivation, triggering extensive metabolic adjustments to sustain available Pi pools, such as an increase in mitochondrial membrane potential, independent of respiration”). We trust that this modification will enhance the interested readers' understanding of our study's overarching conclusions.

**Reviewer #2 (Public Review):**
Summary:Cells cultured in high glucose tend to repress mitochondrial biogenesis and activity, a prevailing phenotype type called Crabree effect that observed in different cell types and cancer. Many signaling pathways have been put forward to explain this effect. Vengayil et al proposed a new mechanism involved in Ubp3/Ubp10 and phosphate that controls the glucose repression of mitochondria. The central hypothesis is that ∆ubp3 shift the glycolysis to trehalose synthesis, therefore lead to the increase of Pi availability in the cytosol, then mitochondrial received more Pi and therefore the glucose repression is reduced.Strengths:The strength is that the authors used an array of different assays to test their hypothesis. Most assays were well designed and controlled.Weaknesses:I think the main conclusions are not strongly supported by the current dataset. Here are my comments on authors' response and model.(1) The authors addressed some of my concerns related to ∆ubp3. But based on the results they observed and discussed, the ∆ubp3 redirect some glycolytic flux to gluconeogenesis while the 0.1% glucose in WT does not. Similarly, the shift of glycolysis to trehalose synthesis is also not relevant to the WT cells cultured in low glucose situation. This should be discussed in the manuscript to make sure readers are not misled to think ∆ubp3 mimic low glucose. It is likely that ∆ubp3 induce proteostasis stress, which is known to activate respiration and trehalose synthesis.But based on the results they observed and discussed, the ∆ubp3 redirect some glycolytic flux to gluconeogenesis while the 0.1% glucose in WT does not. Similarly, the shift of glycolysis to trehalose synthesis is also not relevant to the WT cells cultured in low glucose situation.

We would like to clarify that we do not observe a redirection of glycolytic flux to gluconeogenesis in ubp3 mutant. What we observe is a rewiring of glycolytic flux into increased trehalose synthesis and PPP, and decreased glycolysis. Also, the shift of glycolysis to trehalose synthesis is relevant to WT cells cultured in low glucose. It is a well-known fact that the trehalose synthesis increases with decrease in media glucose. In case of 0.1% glucose, this increase in trehalose is not due to an increase in gluconeogenesis (since the pathways utilizing alternate carbon sources still remain repressed in 0.1% glucose Yin et al., 2003), but by the increase in glycolytic flux towards trehalose. This is also supported by increase in Tps2 protein levels upon decreasing glucose concentration (Shen et al., 2023). We will also note that there are very few studies that actually estimate gluconeogenic flux in cess (and they only rely on steady state measurements). Estimating gluconeogenic flux appropriately is challenging in itself (eg. see Niphadkar et al 2024).

In case of glucose concentrations lower than 0.1%, the shift to trehalose synthesis might not be as relevant. We observe that the glycolysis defective mutant *tdh2tdh3* cells does not show an increase in trehalose synthesis (Figure 3-figure supplement 1E). However, in this context, the decrease in the rate of GAPDH catalyzed reaction alone appears to be sufficient to increase the Pi levels (Figure 3F) even without an increase in trehalose. Therefore, there might be differences in the relative contributions of these two arms towards Pi balance, based on whether it is low glucose in the environment, or a mutant such as *ubp3Δ* that modulates glycolytic flux. In *ubp3Δ* cells, the combination of low rate of GAPDH catalyzed reaction and high trehalose will happen (based on how glycolytic flux is modulated), vs only the low rate of the GAPDH catalyzed reaction in *tdh2tdh3* cells. As an end point the increase in Pi happens in both cases, but this happens via slightly differing outcomes. Also note: in terms of free Pi sources a low-glucose condition (with low glycolytic rate) is very different from a *no-glucose, respiratory condition* (where cells perform very high gluconeogenesis, at a rate that is an order of magnitude higher than in low glucose). In respiration-reliant conditions such as in ethanol, cells switch to high gluconeogenesis, where there is a large increase in trehalose synthesis as a default (eg see Varahan et al 2019). In this condition, trehalose synthesis could become a major source for Pi (eg see Gupta 2021). This could also support the increased mitochondrial respiration. In an ethanol-only medium, the directionality of the GAPDH reaction is itself reversed (i.e. G-1,3-BP → G-3-P). Therefore, this reaction now becomes an added source of Pi, instead of a net consumer of Pi (see illustration in Figure 3G). Therefore, a very reasonable inference is that a combination of increased trehalose and increased 1,3 BPG to G3P conversion can become a Pi source, supporting increased mitochondrial respiration in a non-glucose, respiratory medium.

We have now clarified these points in the discussion section in the updated version of our manuscript. Lines xxx. We hope that this updated discussion section satisfies the reviewer’s concern regarding how relevant the increase in trehalose synthesis is for altered Pi balance and increased mitochondrial respiration in WT cells.

It is likely that ∆ubp3 induce proteostasis stress, which is known to activate respiration and trehalose synthesis.

Apart from some *general* changes in metabolism, there are no reports whatsoever that suggest that general proteostasis stress can results in an extensive, precise metabolic rewiring - where there is an increased in respiration, mitochondrial de-repression, precise decrease in two limiting glycolytic enzyme levels, and a precise reduction in glycolytic flux, as observed in the ubp3 mutant. If this was the case, deletion of *any* deubiquitinase should result in an increase in trehalose and respiration which clearly does not happen (as is already clear from the large screen shown in Figure 1)

However, in response to this query, we performed experiments to assess the extent of proteostasis stress in ubp3 mutants. For this, we have now estimated the changes in global ubiquitination in WT vs ubp3 mutant, and compared this with conditions of moderate proteostasis stress (mild heat shock at 42C/~1hr). These data are now included in the revised manuscript as Figure 1- figure supplement 1J. Notably, our analysis reveals only very minor alteration in global ubiquitination levels in ubp3 mutants compared to WT cells. This is in very stark contrast to limited heat stress, where a clear increase in global ubiquitination can be easily observed. Given these data, we can conclude that there is no significant general proteostatic stress in ubp3 mutants, that could induce substantial metabolic rewiring of such precise nature.

(2) Pi flux: it is known that vacuole can compensate the reduction of Pi in the cytosol. The paper they cited in the response, especially the Van Heerden et al., 2014 showed that the pulse addition of glucose caused transient Pi reduction and then it came back to normal level after 10min or so. If the authors mean the transient change of glycolysis and respiration, they should point that out clearly in the abstract and introduction. If the authors are trying to put out a general model, then the model must be reconsidered.

In Van Heerden et al., the pulse addition of glucose causes transient Pi reduction due to rapid Pi consumption in glycolysis. The phosphate levels came back to normal level because of the glucose flux into trehalose synthesis releasing free Pi. This is the entire crux of the study and this is the reason why tps2 mutants which cannot synthesize trehalose exhibit a growth defect and have decreased Pi levels. As explained in detail in our early response, the cellular Pi levels are maintained by a relative balance of reactions that consume and release Pi and therefore a change in this balance can change Pi as well. Indeed, if this were not the case, the tps2 mutants would simply maintain the Pi levels similar to WT cells by increasing Pi transport from the medium, which is clearly not the case (eg see Gupta 2021).

The cytosol has ~50mM Pi (van Eunen et al., 2010 FEBSJ), while only 1-2mM of glycolysis metabolites, not sure why partial reduction of several glycolysis enzymes will cause significant changes in cytosolic Pi level and make Pi the limiting factor for mitochondrial respiration. In response to this comment, the authors explained the metabolic flux that the rapid, continuous glycolysis will drain the Pi pool even each glycolytic metabolite is only 1-2mM. However, the metabolic flux both consume and release Pi, that's why there is such measurement of overall free Pi concentration amid the active metabolism. One possibility is that the observed cytosolic Pi level changes was caused by the measurement fluctuation.

The measurement fluctuations that we mentioned in our previous response letter was in case of cells grown in high and low glucose, where there are multiple factors such as mitochondrial amount which complicates the Pi measurements. In case of ubp3 mutants which have a similar amount of total mitochondria as that of WT cells, there is minimal fluctuation for Pi measurement. We have done extensive standardization of mitochondrial isolation and Pi measurement in the isolated mitochondria (as explained in detail in the manuscript) to minimize any such fluctuations.

However, the metabolic flux both consume and release Pi, that's why there is such measurement of overall free Pi concentration amid the active metabolism

The reviewer is correct in pointing out that metabolic flux consume and release Pi. However, in glucose grown yeast cells, the rate of glycolysis which is a Pi consuming reaction is higher than any other metabolic pathway. In fact, the glycolytic rate in glucose-grown *S. cerevisiae* is one of the highest ever observed in any living system. A decrease in glycolysis and an increase in trehalose therefore shifts the balance in Pi utilization and results in increased free Pi in ubp3 cells. For a more detailed theoretical reasoning on the consumption and production of Pi, see Gupta 2021.

Importantly, the authors measured Pi inside mito for ethanol and glucose, but not the cytosolic Pi, which is the key hypothesis in their model. The model here is that the glycolysis competes with mito for free cytosolic Pi, so it needs to inhibit glycolysis to free up cytosolic Pi for mitochondrial import to increase respiration. I don't see measurement of cytosolic Pi upon different conditions, only the total Pi or mito Pi. The fact is that in Fig.3C they saw WT+Pi in the medium increase total free Pi more than the ∆ubc3, while WT decrease mito Pi compared to WT control and ∆ubc3 and therefore decrease basal OCR upon Pi supplement. A simple math of Pitotal = Pi cyto + Pi mito tells us that if WT has more Pitotal (Fig.3C) but less Pi mito (fig.5 supp 1C), then it has higher Pi cyto. This is contradictory to what the authors tried to rationalize. Furthermore, as I pointed out previously, the isolated mitochondria can import more Pi when supplemented, so if there is indeed higher Picyto, then the mito in WT should import more Pi. So, to address these contradictory points, the authors must measure Pi in the cytosol, which is a critical experiment not done for their model. For example, they hypothesized that adding 2-DG, or ∆ubp3, suppress glycolysis and thus increase the supply of cytosolic Pi for mito to import, but no cytosolic Pi was measured (need absolute value, not the relative fold changes). It is also important to specific how the experiments are done, was the measurement done shortly after adding 2-DG. Given that the cells response to glucose changes/pulses differently in transient vs stable state, the authors are encouraged to specify that.(1) Importantly, the authors measured Pi inside mito for ethanol and glucose, but not the cytosolic Pi, which is the key hypothesis in their model. The model here is that the glycolysis competes with mito for free cytosolic Pi, so it needs to inhibit glycolysis to free up cytosolic Pi for mitochondrial import to increase respiration. I don't see measurement of cytosolic Pi upon different conditions, only the total Pi or mito Pi.

As clearly described in the manuscript, the key hypothesis that emerges is the role of the availability/accessibility of Pi for the mitochondria, in the context of activity. As discussed in detail in the discussion section, this can come from a combination of available Pi pools in the cytosol *and* increased transport of this Pi to the mitochondria. While it is true that the decreased glycolysis in ubp3 mutants frees up available Pi pools in the cytosol, measurement of cytosolic Pi in these mutants growing in log phase might not necessarily show an increased cytosolic Pi, if the Pi is being actively transported the the mitochondria at a rate higher that the WT, as indicated by the ~6 fold increase in mitochondrial Pi in ubp3 cells. This would require tools such as intracellular fluorescence based-Pi sensors that could accurately capture temporal changes in cytosolic and mitochondrial Pi following glycolytic inhibition. However, these tools are not available till date for use in yeast and measuring cytosolic Pi following glycolytic inhibition over time using colorimetric Pi assays are extremely difficult.

However, the reviewer does correctly state that we had not included measurement of cytosolic Pi. Since the mitochondrial Pi estimate was itself a very challenging (and critical) experiment we had originally thought that data was sufficient. We have therefore now performed a series of new experiments, where we first enrich the cytosolic fraction (without mitochondrial contamination), and estimated cytosolic Pi amounts in WT and ubp3 cells. Our Pi measurements indicate a cytosolic Pi concentration in the range of ~35 mM, which is similar to the earlier reported values in yeast. We further observe that the cytosolic Pi is about ~25% lower in ubp3 mutants (~25-27 mM) compared to WT cells (Figure 4B). As mentioned earlier, this would be consistent with higher transport of Pi from the cytosol to the mitochondria in these cells. Effectively, ubp3 cells have a total increase in cellular Pi, and with a Pi pool distribution such that there is increased Pi availability in mitochondria (Figure 4B). This further substantiates this hypothesis of an increased Pi allocation to mitochondria in ubp3 mutants. The reason for increased rate of Pi transport to mitochondria is not immediately clear, but could also come from changes in cytosolic pH - a possibility that we suggest in our discussion, and is discussed in a later section of this response letter as well.

(2) The fact is that in Fig.3C they saw WT+Pi in the medium increase total free Pi more than the ∆ubc3, while WT decrease mito Pi compared to WT control and ∆ubc3 and therefore decrease basal OCR upon Pi supplement. A simple math of Pitotal = Pi cyto + Pi mito tells us that if WT has more Pitotal (Fig.3C) but less Pi mito (fig.5 supp 1C), then it has higher Pi cyto. This is contradictory to what the authors tried to rationalize. Furthermore, as I pointed out previously, the isolated mitochondria can import more Pi when supplemented, so if there is indeed higher Picyto, then the mito in WT should import more Pi.a) “The fact is that in Fig.3C they saw WT+Pi in the medium increase total free Pi more than the ∆ubc3, while WT decrease mito Pi compared to WT control and ∆ubc3 and therefore decrease basal OCR upon Pi supplement. A simple math of Pitotal = Pi cyto + Pi mito tells us that if WT has more Pitotal (Fig.3C) but less Pi mito (fig.5 supp 1C), then it has higher Pi cyto.”

In WT cells supplemented with external Pi (WT+Pi), there is an increased total Pi, but a decreased mitochondrial Pi. As discussed in the discussion section in the manuscript, this could be due to the supplemented Pi not being transported to mitochondria. The reviewer is correct in pointing out that as per simple math this should mean that the cytosolic Pi in WT+Pi should be high. We have now assessed cytosolic Pi upon external Pi supplementation, and this is exactly what we observe in our cytosolic Pi measurements now included in the revised manuscript (Figure 5-figure supplement 5C). There is a higher cytosolic Pi in WT+Pi (~52 mM) compared to WT cells (~35 mM) and ubp3 cells (~27 mM). We have now pointed this out in the discussion section in the revised manuscript “Notably, this increased respiration does not happen upon direct Pi supplementation to highly glycolytic WT cells, where the Pi accumulates in cytosol, without increasing mitochondrial Pi (Figure 5-figure supplement 1C).” We hope that these new data completely addresses the reviewer’s concern regarding the Pi allocations in case of WT+Pi cells.

b) This is contradictory to what the authors tried to rationalize. Furthermore, as I pointed out previously, the isolated mitochondria can import more Pi when supplemented, so if there is indeed higher Picyto, then the mito in WT should import more Pi.

We would like to clarify that the Pi measurements in WT+Pi absolutely do not contradict our hypothesis. Furthermore, nowhere do we claim that an increase in cytosolic Pi will increase mitochondrial Pi!! On the contrary, we explain in detail that supplementing Pi to WT cells (which increases cytosolic Pi) will not increase respiration if the increased Pi is not being transported to mitochondria. This is exactly what happens in WT+Pi, where Pi accumulates in the cytosol but does not result in increased mitochondrial Pi. The reviewer argues that if there is higher cyto Pi, mitochondria should import more Pi. This is true in case of transport via diffusion where the external concentration dictates the direction of metabolite transport, but is fundamentally wrong in case of transport of metabolites where active transporters and additional regulators are involved. This is the entire basis of the idea of metabolic compartmentalisation where cells maintain pools of metabolites in different organelles which regulate the cellular metabolic state. A well-studied example is pyruvate, whose cytosolic concentration is high in glycolytic cells, but it's transport to mitochondria is reduced in glycolysis to maintain cytosolic fermentation. As discussed in the manuscript, a logical explanation for Pi supplementation not increasing respiration and mitochondria Pi is that there might be mechanisms in highly glycolytic cells that restrict the transport of Pi to mitochondria, thereby compartmentalizing Pi in the cytosol. One such possible mechanism is pH (discussed in a later section) and it is possible that there are other mechanisms involved.

In case of isolated mitochondria, Pi supplementation results in an increased respiration simply because it is an in vitro set up where we supplement metabolites such as pyruvate, malate and ADP along with phosphate to ensure that mitochondria is actively respiring and in this case Pi will be consumed since it is being used for ATP synthesis. This is entirely different from an in vivo scenario where cells are glycolytic, and mechanisms to prevent mitochondrial transport of metabolites such as pyruvate and phosphate are active.

c) It is also important to specific how the experiments are done, was the measurement done shortly after adding 2-DG?

Cells were treated with 2-DG for one hour and respiration was measured. We have mentioned these details clearly in the figure legends and methods.

d) The most likely model to me is that, which is also the consensus in the field, is that no matter 2-DG or ∆ubp3, the cells re-wiring metabolism in both cytosol and mitochondria, and it is the total network shift that cause the mitochondrial respiration increase, which requires the increase of mito import of Pi, ADP, O2, and substrates, but not caused/controlled by the Pi that singled out by the authors in their model.

The aim of our study is only to highlight the importance of mitochondrial Pi availability as a critical factor in controlling mitochondrial respiration. Of course this would require sufficient other factors such as ADP, substrates and oxygen. It cannot be otherwise. However, as we point out in the discussion, a major limiting factor might be Pi availability. While the altered glycolysis in ubp3 mutants might control availability of other factors such as pyruvate and ADP, this is not the focus of our study. We would also like to point out that prior studies show that even though cytosolic ADP decreases in the presence of glucose, this does not limit mitochondrial ADP uptake, or decrease respiration, due to the very high affinity of the mitochondrial ADP transporter. This is discussed in our discussion section as well. Further we show that the levels of ETC proteins can be altered by changing Pi levels, which places Pi as a major regulator of respiration. We would like to point out once again that studies in other systems have also highlighted a major role of mitochondrial Pi availability in controlling respiration. These references are included in our manuscript (Scheibye-Knudsen et al., 2009, Seifer et al., 2015). This includes a recent study in T cells that clearly shows increased mitochondrial respiration upon overexpressing mitochondrial Pi transporter SLC25A3 alone (Wu et al., 2023). Our manuscript now in fact provides a contextual explanation of these diverse observations from other cellular systems where mitochondrial Pi transport appears to regulate respiration.

(3) The explanation that cytosolic pH reduction upon glucose depletion/2DG is a mistake. There are a lot of data in the literature showing the opposite. If the authors do think this is true, then need to show the data. Again, it is important to distinguish transient vs stable state for pH changes.

We observe that directly supplementing Pi to WT cells growing in high glucose does not result in higher mitochondrial Pi or increased respiration. However, supplementing Pi to WT cells increases mitochondrial respiration in the presence of glycolytic inhibitor 2-DG. We therefore merely suggest that cytosolic pH could be an additional regulator of mitochondrial Pi transport, since this will be consistent with the differences in mitochondrial Pi transport in highly glycolytic cells, and cells with decreased glycolysis (such as 2-DG addition and ubp3 mutant). This is because in mitochondria, Pi is co-transported along with protons. Therefore, changes in cytosolic pH (which changes the proton gradient) will control the mitochondrial Pi transport (Hamel et al., 2004). The glycolytic rate is itself a major factor that controls cytosolic pH. The cytosolic pH in highly glycolytic cells is maintained ~7, and decreasing glycolysis results in cytosolic acidification (Orij et al., 2011). Therefore, under conditions of decreased glycolysis (such as loss of Ubp3), cytosolic pH becomes acidic. Since mitochondrial Pi transport depends on the proton gradient, a low cytosolic pH would favour mitochondrial Pi transport. Therefore, under conditions of decreased glycolysis (2DG treatment, or loss of Ubp3), where cytosolic pH would be acidic, increasing cytosolic Pi might indirectly increase mitochondria Pi transport, thereby leading to increased respiration. But we certainly do leave alternate interpretations to the imagination of any reader, and are indeed open to them. These are all exciting future directions this study will enable a contextual interpretation of.

The explanation that cytosolic pH reduction upon glucose depletion/2DG is a mistake.

We have cited two independent studies which suggest that cytosolic pH decreases upon a decrease in glycolysis (Orij et al.,2011 ,Dechant et al., 2010). This control of cytosolic pH by the glycolytic rate has been extensively shown using glycolytic mutants, cells in low glucose and cells grown in the presence of glycolytic inhibitors. According to the reviewer, this is a mistake and

there are a lot of data in the literature showing the opposite.

In our literature review we did not come across any relevant studies that actually show the opposite. If the reviewer still thinks this is a mistake, the reviewer is welcome to include some of the relevant literature that clearly shows the opposite in the comments, with actual measurements of cytosolic pH. Additionally, the possible role of cytosolic pH in this context does not affect the conclusions of our study, and we only include this as a possibility in the discussion. Therefore, this is obviously well beyond the scope of experiments in our current study, and considering the extensive data from multiple studies that shows that cytosolic pH decreases under low glycolysis, there is no relevance to including experiments to address the same in this study. We leave this as a point for an interested reader to think about, and it certainly can nucleate new directions of future study.